# A Consensus Anchor-Guided Hypergraph Framework for Incomplete Multi-View Clustering

## Abstract

As a significant task within the field of unsupervised learning, Incomplete Multi-View Clustering (IMVC) faces considerable challenges in scenarios involving large-scale datasets, heterogeneous data, and missing views. Existing anchor-based clustering approaches primarily reduce computational and storage overhead by introducing anchors, yet they often focus on binary sample-anchor relationships. These methods lack robust learning of consensus anchors under missing conditions and fail to effectively model high-order relationships among samples. Furthermore, systematic discussions regarding implementation details and robustness mechanisms remain insufficient. To address this, this paper proposes a Missing-aware Consensus Anchor-guided Hypergraph Clustering (MCAHC) framework. This method constructs hypergraph through sample-anchor connections and anchor guidance to capture high-order relationships among samples, effectively mitigating view-missing and noise interference. Concurrently, it designs sample-level and view-level reweighting mechanisms to suppress inter-view imbalance and promote cross-view consistency, while explicitly down-weighting severely incomplete samples to prevent them from biasing anchor selection. Experimental results demonstrate that MCAHC provides an efficient and robust solution for multi-view clustering in large-scale and high-missing-value scenarios.

## 1 Introduction

Incomplete Multi-View Clustering (IMVC) aims to partition incomplete data into semantically consistent clusters by exploiting information from multiple heterogeneous views(Lin et al. (2021);Wen et al. (2023);Xu et al. (2024)). In many applications, the complementarity among views can significantly improve clustering performance. However, some challenges remain insufficiently addressed in practice.

Some samples are missing features in certain views, which invalidates the traditional assumption that each sample has observations in all views. A number of works have been proposed to tackle this issue. Wen et al. (2024) introduced a diffusion-based framework for missing-view generation, combined with data augmentation strategies to improve clustering under high missing rates. Chao et al. (2024) developed an contrastive learning framework that jointly optimizes missing-view handling, representation learning, and clustering assignment via graph consistency transfer, instance-level attention, and high-confidence guidance. Yu et al. (2025b) proposed a simple yet effective method, which performs similarity-level imputation and introduces hybrid prototype groups for each view, thereby enhancing multi-scale similarity modeling and clustering performance within a unified framework. Additionally, another clustering methods simplifies graph structure construction by utilizing anchors, thereby reducing computational overhead while balancing efficiency and effectiveness. Such anchor-based clustering approaches offer advantages such as scalability, reduced memory for graph storage, and improved stability. For instance, Zhang et al. (2024) propose a cluster structure regularization method that simultaneously optimizes anchor and cluster assignments, making anchors adaptive and more discriminative while balancing efficiency and accuracy. Liu et al. (2024a) systematically review anchor generation and anchor map construction workflows, proposing plug-and-play anchor enhancement strategies that leverage cross-view correlations to strengthen anchor maps and improve multi-view fusion performance. Zhang et al. (2025) demonstrate that an-

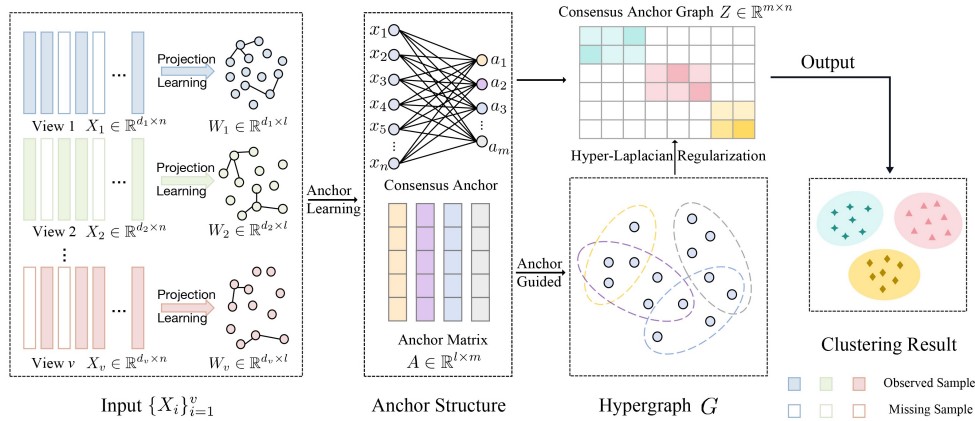

Figure 1: Overview of the proposed MCAHC framework for incomplete multi-view clustering. Incomplete multi-view input set $\{X_i\}_{i=1}^v$ is projected onto a low-dimensional representation via the projection set $\{W_i\}_{i=1}^v$. Consensus anchor learning then produces the anchor $A \in \mathbb{R}^{l \times m}$ and the anchor graph $Z \in \mathbb{R}^{m \times n}$. Each anchor induces a hyperedge, and samples are connected to their most similar anchors, forming an anchor-guided hypergraph $G$ with connected samples. The hypergraph Laplacian regularization term captures high-order information, leading to robust clustering results.

chors significantly reduce computational complexity in large-scale scenarios and propose an anchor-aware representation learning scheme to model latent relationships between anchors while maintaining scalability.

Despite progress along these two directions, existing methods still face limitations in real-world scenarios. For incomplete multi-view clustering, many methods assume that view-missingness is random or balanced, or that observed samples are evenly distributed across views, which rarely holds in practice(Xu et al. (2024);Han et al. (2024)). Missing samples often exhibit distributional shifts compared to complete ones, leading to bias. Furthermore, methods relying on view recovery or similarity-graph construction may introduce noise during imputation or estimation, which can propagate and deteriorate clustering performance. For anchor-based approaches, anchor selection or generation is often fixed or heuristic, making them less adaptable to structural differences across views. Moreover, most methods only consider pairwise relations between anchors and samples, while neglecting high-order relations involving groups of samples and anchors(Li et al. (2022);Mei et al. (2024);Chen et al. (2025)).

To overcome these limitations, we introduce a missing-aware mechanism that adaptively adjusts the contributions of incomplete samples across views via sample-level masks and weighting matrices. In addition, we propose an anchor-guided hypergraph framework for multi-view clustering. In our design, each anchor induces a hyperedge, and samples are connected to their most similar anchors according to similarity scores, naturally forming an anchor-sample hypergraph that captures high-order relations beyond pairwise connections. We further incorporate a hypergraph Laplacian regularization term to enforce cross-view structural consistency while preserving the scalability benefits of anchors. This yields a unified model capable of handling incomplete multi-view data.

The main contributions of this paper are summarized as follows:

- Unlike anchor-based methods that only encode pairwise relations, we propose a **anchor-guided hypergraph Laplacian regularization term**, which elevates bipartite anchor graphs into high-order structures to better capture anchor–sample group interactions.

- We incorporate a **missing-aware mechanism** that performs **sample-level and view-level reweighting**, not only alleviating inter-view imbalance and enhancing cross-view consistency, but also preventing severely missing data points from dominating **anchor selection**.

- We propose an **alternating optimization algorithm** and provide detailed derivations of its update rules. Experiments demonstrate that it achieves strong performance across diverse datasets and missing-rate regimes while **significantly improving efficiency**.

## 2 RELATED WORK

In multi-view clustering, directly constructing similarity graphs in the original high-dimensional feature space is computationally prohibitive and difficult to scale(He et al. (2025);Liu et al. (2024b);Wang et al. (2022b);Yu et al. (2025a)). To tackle this, the basic anchor graph model introduces view-specific projection matrices to map original data into a shared low-dimensional latent space, while employing a small set of representative anchors to approximate the entire sample set(Chen et al. (2024);Sun et al. (2021);Wang et al. (2022a);Qin et al. (2025);Qin et al. (2025)). This joint modeling of projection and anchors effectively reduces computational complexity while preserving essential structural information, and has thus become the cornerstone for subsequent methodological advances.

Given $v$ views $\{\mathbf{X}_p\}_{p=1}^{v}$ with $\mathbf{X}_p \in \mathbb{R}^{d_p \times n}$, let $\mathbf{W}_p \in \mathbb{R}^{d_p \times l}$ be view projections to a $l$-dimensional consensus space, $\mathbf{A} \in \mathbb{R}^{l \times m}$ the shared anchor matrix, $\mathbf{Z} \in \mathbb{R}^{m \times n}$ the anchor graph, and $\beta_p \geq 0$ the view weights with $\sum_{p=1}^{v} \beta_p = 1$. The model can be written as

$$\min_{\{\mathbf{W}_p\},\, \mathbf{A},\, \mathbf{Z},\, \{\beta_p\}} \sum_{p=1}^{v} \beta_p^2 \left\| \mathbf{X}_p - \mathbf{W}_p \mathbf{A} \mathbf{Z} \right\|_F^2 \;+\; \left\| \mathbf{Z} \right\|_F^2$$

$$\text{s.t.} \quad \mathbf{W}_p^\top \mathbf{W}_p = \mathbf{I}, \;\; \mathbf{A}^\top \mathbf{A} = \mathbf{I}, \;\; \mathbf{Z} \geq 0, \;\; \mathbf{Z}^\top \mathbf{1} = \mathbf{1}, \;\; \sum_{p=1}^{v} \beta_p = 1. \tag{1}$$

Clustering is then performed on the consensus graph derived from $\mathbf{Z}$.

Following this paradigm, Wang et al. (2022a) first introduced the anchor graph framework into incomplete multi-view clustering, where unified anchor learning and incomplete anchor graph construction are combined to form a consensus anchor graph, thereby maintaining cross-view structural consistency and alleviating the high complexity of large-scale IMVC. Liu et al. (2022) unified anchor learning and graph construction within a single framework, further imposing connectivity constraints to directly generate graphs with precise cluster structures, enabling one-step clustering results without additional post-processing or hyperparameter tuning. Chen et al. (2024) enhanced the classical anchor graph framework by introducing an index matrix to naturally handle both complete and incomplete data, and by stacking anchor graphs from multiple views into a tensor with low-rank constraints to explicitly capture high-order cross-view correlations. Ou et al. (2024) proposed hierarchical feature descent within the anchor model, mapping views of varying dimensionalities into a unified subspace, and then learning a shared anchor matrix and consensus bipartite graph to alleviate view discrepancy and improve scalability. Qin et al. (2025) further integrated graph construction, anchor learning, and graph partition into a unified framework where the three components reinforce each other; by learning a shared anchor graph to ensure cross-view consistency and explicitly linking it with symmetric nonnegative matrix factorization, the clustering results can be directly obtained.

## 3 METHODOLOGY

Building upon the baseline anchor graph formulation in model (1), we develop a novel missing-aware anchor-guided hypergraph multi-view clustering framework. Specifically, we extend the conventional bipartite anchor graph into a hypergraph structure to capture high-order relations (Section 3.1), introduce a missing-aware weighting mechanism to adaptively handle incomplete data (Section 3.2), and finally integrate these components into a unified framework (Section 3.3).

### 3.1 ANCHOR-GUIDED HYPERGRAPH

We build an anchor-guided hypergraph $\mathcal{H} = (\mathcal{V}, \mathcal{E})$ where vertices contain both samples and anchors, $\mathcal{V} = \mathcal{V}_s \cup \mathcal{V}_a$ with $|\mathcal{V}_s| = n$ and $|\mathcal{V}_a| = m$. Each anchor induces exactly one hyperedge, hence $|\mathcal{E}| = m$ and $\mathcal{E} = \{e_1, \ldots, e_m\}$, where $e_j$ aggregates samples that are similar to anchor $j$.

The weighted incidence matrix $\mathbf{H} \in \mathbb{R}^{(n+m) \times m}$ represents the hypergraph over $n$ samples and $m$ anchors. Let $\mathbf{Z} \in \mathbb{R}^{m \times n}$ denote the anchor graph, where each entry $z_{j,i} \geq 0$ measures the similarity between sample $x_i$ and anchor $a_j$. For each sample $i \in \{1, \ldots, n\}$, we identify the index

set $\mathcal{N}_T(i) \subseteq \{1, \ldots, m\}, (1 \leq T \leq m)$, corresponding to the $T$ anchors with the largest similarity scores $z_{j,i}$. The sample–to–hyperedge incidences are then defined as

$$\mathbf{H}_{i,j} = \begin{cases} z_{j,i}, & j \in \mathcal{N}_T(i), \\ 0, & \text{otherwise.} \end{cases} \tag{2}$$

For the anchor rows, each anchor is associated with a dedicated hyperedge. Specifically, for $j = 1, \ldots, m$, we set $\mathbf{H}_{n+j, j} = 1, \mathbf{H}_{n+j, \ell} = 0 (\ell \neq j)$. Thus, each hyperedge consists of its $T$ most similar samples, weighted by their similarity scores $z_{j,i}$, together with the anchor $a_j$ itself, which is included with unit weight.

Define vertex degrees $d(v) = \sum_e \mathbf{H}_{ve}$ and edge degrees $\delta(e) = \sum_v \mathbf{H}_{ve}$. Let $\mathbf{D}_v = \mathrm{Diag}(d(v)) \in \mathbb{R}^{(n+m) \times (n+m)}$ and $\mathbf{D}_e = \mathrm{Diag}(\delta(e)) \in \mathbb{R}^{m \times m}$. Using unit hyperedge weights, the normalized hypergraph Laplacian is

$$\mathbf{L}_H = \mathbf{I} - \mathbf{D}_v^{-1/2} \mathbf{H} \mathbf{D}_e^{-1} \mathbf{H}^\top \mathbf{D}_v^{-1/2} \in \mathbb{R}^{(n+m) \times (n+m)}. \tag{3}$$

We form a sample embedding $\mathbf{Z}_s = \mathbf{A}\mathbf{Z} \in \mathbb{R}^{l \times n}$ and stack it with the anchor embedding $\mathbf{A} \in \mathbb{R}^{l \times m}$ to obtain $\mathbf{Z}_{\mathrm{aug}} = [\mathbf{Z}_s \ \mathbf{A}] \in \mathbb{R}^{l \times (n+m)}$, so that samples and anchors lie in the same latent space and are jointly regularized by $\mathbf{L}_H$ through a Laplacian regularization term $\mathrm{Tr}(\mathbf{Z}_{\mathrm{aug}} \mathbf{L}_H \mathbf{Z}_{\mathrm{aug}}^\top)$.

### 3.2 Missing-aware Framework

Let $\mathbf{S}_p = \mathrm{Diag}(s_p^{(i)}) \in \mathbb{R}^{n \times n}$ be a per-view diagonal mask matrix with $s_p^{(i)} \in \{0, 1\}$ indicating whether sample $i$ is observed in view $p$. We define the completeness weight of view $p$ as

$$\alpha_p = \frac{\sum_{i=1}^n s_p^{(i)}}{\sum_{u=1}^v \sum_{i=1}^n s_u^{(i)}}, \qquad \sum_{p=1}^v \alpha_p = 1, \tag{4}$$

and the missingness of sample $i$ by $q_i = 1 - \sum_{p=1}^v \alpha_p s_p^{(i)} \in [0, 1]$. Let $\mathbf{Q} = \mathrm{Diag}(q_i)$ be the missing-rate regularizer. To downweight highly-missing samples during reconstruction, we use adaptive sample weights $m_i = e^{-\gamma q_i}, \mathbf{M} = \mathrm{Diag}(m_i)$. Intuitively, the larger $q_i$, the less reliable the sample, hence the smaller $m_i$, meanwhile the penalty term $\|\mathbf{Z}\mathbf{Q}^{1/2}\|_F^2$ discourages anchor assignments that rely on highly-missing samples.

### 3.3 Our Framework

Overall, the object function can be written as

$$\min_{\beta, \{\mathbf{W}_p\}, \mathbf{A}, \mathbf{Z}} \sum_{p=1}^v \beta_p^2 \left\| (\mathbf{W}_p^\top \mathbf{X}_p - \mathbf{A}\mathbf{Z}) \mathbf{S}_p \mathbf{M}^{1/2} \right\|_F^2 + \lambda_1 \mathrm{Tr}(\mathbf{Z}_{\mathrm{aug}} \mathbf{L}_H \mathbf{Z}_{\mathrm{aug}}^\top) + \lambda_2 \|\mathbf{Z}\mathbf{Q}^{1/2}\|_F^2$$

$$s.t. \quad \beta_p \geq 0, \sum_{p=1}^v \beta_p = 1; \quad \mathbf{W}_p^\top \mathbf{W}_p = \mathbf{I}; \quad \mathbf{A}^\top \mathbf{A} = \mathbf{I}; \quad \mathbf{Z} \geq 0, \mathbf{Z}^\top \mathbf{1} = \mathbf{1}. \tag{5}$$

The first term aligns multi-view reconstructions to the shared anchor embedding, masked by $\mathbf{S}_p$ and reweighted by $\mathbf{M}$. The second term imposes hypergraph regularization on both samples $\mathbf{Z}_s$ and anchors $\mathbf{A}$ through $\mathbf{Z}_{\mathrm{aug}}$ and $\mathbf{L}_H$. The third penalty term primarily serves to reduce the impact of samples with high missing rates on anchor quality.

## 4 Optimization

We design an alternating algorithm for optimizing each variable in Eq.(5) by fixing the others.

**Updating $\mathbf{W}_p$:** With other variables fixed, the $p$-th view subproblem reduces to

$$\max_{\mathbf{W}_p^\top \mathbf{W}_p = \mathbf{I}} \mathrm{Tr}(\mathbf{W}_p^\top \mathbf{G}_p), \ \mathbf{G}_p = \mathbf{X}_p \mathbf{S}_p \mathbf{M} \mathbf{Z}^\top \mathbf{A}^\top. \tag{6}$$

Let the SVD be $\mathbf{G}_p = \mathbf{U}_p \mathbf{\Sigma}_p \mathbf{V}_p^\top$. The optimum is $\mathbf{W}_p^\star = \mathbf{U}_p \mathbf{V}_p^\top$.

---

**Algorithm 1 Z-update Algorithm**

---

**Input:** $\mathbf{K} \succeq 0 \in \mathbb{R}^{n \times n}$, $\mathbf{J} \in \mathbb{R}^{m \times n}$; initial $\mathbf{Z}^{(0)} = \mathbf{Z}^{(1)}$, $t_1 = 1$; stepsize $L \geq 2\lambda_{\max}(\mathbf{K})$; tolerance $\varepsilon$.

1: **while** not converged **do**
2:     $t_{t+1} \leftarrow \left(1 + \sqrt{1 + 4t_t^2}\right)/2$
3:     $\mathbf{Y}^{(t)} \leftarrow \mathbf{Z}^{(t)} + \left(t_t - 1/t_{t+1}\right)\left(\mathbf{Z}^{(t)} - \mathbf{Z}^{(t-1)}\right)$
4:     $\widetilde{\mathbf{Z}} \leftarrow \mathbf{Y}^{(t)} - L^{-1}\left(2\mathbf{Y}^{(t)}\mathbf{K} - 2\mathbf{J}\right)$
5:     **for each** column $j = 1, \ldots, n$ **do**    $\mathbf{Z}_{:,j}^{(t+1)} \leftarrow \Pi_\Delta\left(\widetilde{\mathbf{Z}}_{:,j}\right)$, where $\Delta = \{z \in \mathbb{R}^m \mid z \geq 0,\ \mathbf{1}^\top z = 1\}$
6:     **if** $\dfrac{\|\mathbf{Z}^{(t+1)} - \mathbf{Z}^{(t)}\|_F}{\max\{1, \|\mathbf{Z}^{(t)}\|_F\}} < \varepsilon$ **then break**
7: **end while**
**Output:** Updated $\mathbf{Z}$

---

**Updating A:** With other variables fixed and discarding constants gives

$$\max_{\mathbf{A}^\top \mathbf{A} = \mathbf{I}} \operatorname{Tr}\left(\mathbf{A}^\top \mathbf{P}\right),\ \mathbf{P} = \sum_{p=1}^{v} \beta_p^2 \mathbf{W}_p^\top \mathbf{X}_p \mathbf{S}_p \mathbf{M} \mathbf{Z}^\top. \tag{7}$$

Let $\mathbf{P} = \mathbf{U}\boldsymbol{\Sigma}\mathbf{V}^\top$. Then $\mathbf{A}^\star = \mathbf{U}\mathbf{V}^\top$.

**Updating Z:** Block-partitioned as $\mathbf{L}_H = \begin{bmatrix} \mathbf{L}_{dd} & \mathbf{L}_{da} \\ \mathbf{L}_{ad} & \mathbf{L}_{aa} \end{bmatrix}$ with $\mathbf{L}_{ad} = \mathbf{L}_{da}^\top$. Using $\mathbf{Z}_{\text{aug}} = [\mathbf{Z}_s\ \mathbf{A}]$ and the block form of $\mathbf{L}_H$, we obtain

$$\operatorname{Tr}(\mathbf{Z}_{\text{aug}}\mathbf{L}_H\mathbf{Z}_{\text{aug}}^\top) = \operatorname{Tr}(\mathbf{Z}\mathbf{L}_{dd}\mathbf{Z}^\top) + 2\operatorname{Tr}(\mathbf{Z}^\top\mathbf{L}_{ad}) + \operatorname{Tr}(\mathbf{A}\mathbf{L}_{aa}\mathbf{A}^\top), \tag{8}$$

where the last term is constant. Collecting the quadratic and linear terms, the **Z**-subproblem becomes

$$\min_{\mathbf{Z} \in \mathbb{R}^{m \times n}} f(\mathbf{Z}) = \operatorname{Tr}(\mathbf{Z}\mathbf{K}\mathbf{Z}^\top) - 2\operatorname{Tr}(\mathbf{Z}^\top\mathbf{J}) \quad \text{s.t.} \quad \mathbf{Z} \geq 0,\ \mathbf{Z}^\top\mathbf{1} = \mathbf{1}, \tag{9}$$

with $\mathbf{K} = \sum_{p=1}^{v} \beta_p^2 \mathbf{S}_p \mathbf{M} + \lambda_1 \mathbf{L}_{dd} + \lambda_2 \mathbf{Q} \succeq 0$, $\mathbf{J} = \sum_{p=1}^{v} \beta_p^2 \mathbf{A}^\top \mathbf{W}_p^\top \mathbf{X}_p \mathbf{S}_p \mathbf{M} - \lambda_1 \mathbf{L}_{ad}$. We solve Eq.(9) by FISTA(Beck & Teboulle (2009)) with gradient $\nabla f(\mathbf{Z}) = 2\mathbf{Z}\mathbf{K} - 2\mathbf{J}$ and stepsize $L \geq 2\lambda_{\max}(\mathbf{K})$, followed by column-wise Euclidean projection onto the probability simplex $\Delta = \{z \in \mathbb{R}^m : z \geq 0,\ \mathbf{1}^\top z = 1\}$; see Algorithm 1.

**Updating $\boldsymbol{\beta}$:** With other variables being fixed, the objective function for $\beta_p$ is

$$\min_{\boldsymbol{\beta} \geq 0, \sum \beta_p = 1} \sum_p \beta_p^2 R_p^2, \tag{10}$$

where $R_p = \left\|\left(\mathbf{W}_p^\top \mathbf{X}_p - \mathbf{A}\mathbf{Z}\right)\mathbf{S}_p\mathbf{M}^{1/2}\right\|_F$. We can obtain the optimal based on Cauchy-Schwarz inequality as $\beta_p^\star = \dfrac{R_p^{-1}}{\sum_{u=1}^{v} R_u^{-1}}$.

**Updating $\mathbf{L}_H$:** We rebuild the anchor-guided hypergraph from the current embeddings. For each sample $i$, form its latent code $z_i$ (the $i$-th column of $\mathbf{Z}$); connect $i$ to its $T$ most similar anchors with the largest $z_{ji}$. Construct the incidence matrix $\mathbf{H} \in \{0,1\}^{(n+m) \times m}$. Then update the normalized Laplacian

$$\mathbf{L}_H = \mathbf{I} - \mathbf{D}_v^{-1/2}\mathbf{H}\mathbf{D}_e^{-1}\mathbf{H}^\top\mathbf{D}_v^{-1/2}, \tag{11}$$

where $\mathbf{D}_v$ and $\mathbf{D}_e$ are vertex and edge degree diagonals.

We summarize the overall procedure for solving the optimization problem (2) in Algorithm 2.

---

**Algorithm 2** Alternating optimization for the proposed model

---

**Input:** $\{\mathbf{X}_p, \mathbf{S}_p\}_{p=1}^v$, diagonal $\mathbf{M}, \mathbf{Q}$, anchor number $m$, cluster number $k$; $\lambda_1, \lambda_2$; $T$ for hypergraph.
1: **repeat**
2:  Update $\{\mathbf{W}_p\}_{p=1}^v$ by solving (6);
3:  Update $\mathbf{A}$ by solving (7);
4:  Update $\mathbf{Z}$ by Algorithm 1;
5:  Update $\{\beta_p\}_{p=1}^v$ by solving (10);
6:  Update $\mathbf{L}_H$ by equation (11)
7: **until** relative change $< \varepsilon$;
**Output:** Cluster labels from $k$-means on $\mathbf{U}_k$ (derived via SVD of $\mathbf{Z}$)

---

## 5 EXPERIMENTS

In this section, we evaluate MCAHC on six multi-view datasets under three missingness levels and compare it against eight representative baselines. We also report running-time comparisons, ablation studies, convergence analysis, and parameter-sensitivity analyses to demonstrate the model's effectiveness and efficiency.

### 5.1 BASELINES AND DATASETS

We conduct experiments on six multi-view datasets, with specific details provided in Table 1.

Table 1: General Statistics of Datasets

| Dataset | Sample | View | Class | Feature Dimension |
|---|---|---|---|---|
| NGs | 500 | 3 | 5 | 2000/2000/2000 |
| Caltech101-20 | 2396 | 6 | 20 | 48/40/254/1984/512/928 |
| BDGP | 2500 | 3 | 5 | 1000/500/250 |
| CCV | 6773 | 3 | 20 | 20/20/20 |
| Animal | 11673 | 4 | 50 | 2689/2000/2001/2000 |
| MNIST | 60000 | 3 | 10 | 342/1024/64 |

MCAHC is compared with the following multi-view clustering approaches: non-anchor-based clustering methods (**BSV**(Ng et al. (2001));**HCP-IMSC**(Li et al. (2022));**UOMVSC**(Tang et al. (2023));**SCSL**(Liu et al. (2024c))) and anchor-based clustering methods (**EMKMC**(Yang et al. (2023));**FastMICE**(Huang et al. (2023));**FDAGF**)(Zhang et al. (2023));**MVSC-HFD**(Ou et al. (2024))).

### 5.2 RESULTS AND DISCUSSIONS

We employ three widely adopted metrics to evaluate clustering results: Accuracy (ACC), Normalized Mutual Information (NMI), and Purity (PUR). To mitigate randomness, each experiment is repeated 20 times, with the mean and variance reported. Specifically, Tables 2 and 3 present the clustering results for all multi-view clustering methods under ACC, NMI, and PUR metrics at missing rates of 30%, 50%, and 70%. Methods unable to compute on a dataset due to insufficient memory are denoted as N/A. Based on the clustering results obtained from Tables 2 and 3, we draw the following conclusions:

- MCAHC outperformed most comparison algorithms under various missing rates and evaluation metrics. For instance, it consistently achieved the best performance on the NGs and BDGP datasets, while yielding second-best results on Caltech101-20 and Animal datasets. Even with a missing rate as high as 70%, MCAHC demonstrated satisfactory performance across three metrics. This demonstrates that MCAHC effectively addresses the IMVC task.

Table 2: Clustering Results on Datasets

| Method | NGs | | | | | | | | |
|---|---|---|---|---|---|---|---|---|---|
| | **30%** | | | **50%** | | | **70%** | | |
| | ACC | NMI | PUR | ACC | NMI | PUR | ACC | NMI | PUR |
| BSV | 39.07±1.58 | 19.43±0.87 | 39.68±1.60 | 33.15±1.42 | 13.80±1.39 | 34.03±1.14 | 25.74±0.83 | 6.71±0.75 | 26.28±0.85 |
| HCP-IMSC | 93.40±0.00 | 80.31±0.00 | 93.40±0.00 | 89.00±0.00 | 71.46±0.00 | 89.00±0.00 | 85.10±0.00 | 60.49±0.00 | 85.10±0.00 |
| SCSL | 60.77±0.73 | 42.52±0.10 | 64.60±0.22 | 38.72±0.65 | 39.92±0.22 | 36.86±0.19 | 29.11±0.16 | 38.31±0.11 | 30.18±0.14 |
| UOMVSC | 73.17±0.01 | 67.02±0.00 | 73.93±0.00 | 73.04±0.01 | 65.81±0.00 | 72.88±0.00 | 71.11±0.00 | 60.84±0.00 | 70.74±0.00 |
| EMKMC | 45.13±0.00 | 38.01±0.00 | 45.15±0.00 | 44.78±0.00 | 37.62±0.00 | 44.76±0.00 | 42.53±0.00 | 34.57±0.00 | 41.30±0.00 |
| FastMICE | 40.42±0.03 | 18.23±0.09 | 41.37±0.08 | 39.23±0.05 | 16.34±0.07 | 40.41±0.07 | 37.23±0.03 | 14.25±0.17 | 38.23±0.07 |
| FDAGF | 53.33±0.00 | 34.75±0.00 | 54.99±0.00 | 52.93±0.00 | 33.82±0.00 | 54.10±0.00 | 52.82±0.00 | 33.21±0.00 | 54.32±0.00 |
| MVSC-HFD | 46.76±6.46 | 24.01±5.77 | 47.88±6.24 | 42.40±2.78 | 17.87±2.62 | 42.92±3.21 | 37.80±2.62 | 12.38±1.27 | 38.12±2.80 |
| **Ours** | **94.20±0.00** | **83.53±0.00** | **94.20±0.00** | **91.20±0.00** | **77.38±0.00** | **91.20±0.00** | **87.42±0.09** | **69.37±0.13** | **87.42±0.09** |

| Method | Caltech101-20 | | | | | | | | |
|---|---|---|---|---|---|---|---|---|---|
| | **30%** | | | **50%** | | | **70%** | | |
| | ACC | NMI | PUR | ACC | NMI | PUR | ACC | NMI | PUR |
| BSV | 39.71±3.14 | 53.04±1.05 | 68.77±1.13 | 36.86±3.84 | 49.04±1.68 | 65.32±1.46 | 33.01±2.75 | 43.96±1.06 | 61.05±1.11 |
| HCP-IMSC | 46.44±2.21 | 50.38±0.99 | 66.76±0.59 | 42.66±1.82 | 50.97±0.96 | 67.07±0.92 | 41.13±1.48 | 50.50±0.66 | 66.42±0.69 |
| SCSL | 43.84±1.66 | 57.30±0.68 | 75.50±0.61 | 43.39±1.83 | 55.83±0.57 | 72.78±0.84 | 45.05±1.64 | 52.81±0.73 | 70.04±0.77 |
| UOMVSC | 44.98±0.01 | **60.17±0.00** | 75.68±0.07 | 41.79±0.00 | **57.48±0.01** | 72.63±0.00 | 38.57±0.00 | 53.79±0.00 | 68.57±0.02 |
| EMKMC | 30.87±0.00 | 32.64±0.00 | 56.12±0.00 | 28.57±0.00 | 31.47±0.00 | 54.01±0.00 | 27.50±0.00 | 31.02±0.00 | 53.45±0.00 |
| FastMICE | 34.27±2.02 | 59.32±0.86 | 75.24±0.57 | 33.50±1.42 | 57.23±0.31 | 73.25±0.52 | 34.50±1.22 | 53.34±0.23 | 70.26±0.84 |
| FDAGF | 41.22±2.46 | 49.23±0.07 | 67.25±2.61 | 43.12±2.56 | 50.15±0.00 | 69.36±1.96 | 40.49±3.72 | 48.13±0.02 | 66..20±2.84 |
| MVSC-HFD | 51.09±3.17 | 45.63±1.94 | 64.69±1.79 | 48.01±2.53 | 42.81±1.58 | 63.10±1.81 | 41.94±3.88 | 38.25±1.61 | 58.18±2.33 |
| **Ours** | **55.23±1.99** | 59.81±0.64 | **76.34±0.44** | **55.34±2.42** | 57.27±0.60 | **73.34±0.42** | **51.53±1.99** | **55.91±0.45** | **73.72±0.42** |

| Method | BDGP | | | | | | | | |
|---|---|---|---|---|---|---|---|---|---|
| | **30%** | | | **50%** | | | **70%** | | |
| | ACC | NMI | PUR | ACC | NMI | PUR | ACC | NMI | PUR |
| BSV | 36.22±0.85 | 21.40±0.92 | 38.02±0.92 | 32.88±0.66 | 16.76±0.71 | 33.82±0.69 | 31.42±0.69 | 14.77±0.75 | 32.69±0.71 |
| HCP-IMSC | 34.28±0.36 | 12.76±0.02 | 36.38±0.01 | 32.44±0.20 | 12.37±0.04 | 35.35±0.01 | 33.25±0.04 | 11.72±0.02 | 34.75±0.02 |
| SCSL | 29.08±0.89 | 9.19±2.69 | 30.28±0.11 | 30.89±1.96 | 6.72±1.94 | 31.29±1.96 | 29.69±1.88 | 4.71±2.54 | 30.09±1.88 |
| UOMVSC | 38.97±0.03 | 15.56±0.00 | 41.69±0.01 | 36.34±0.00 | 14.24±0.01 | 39.36±0.04 | 33.29±0.00 | 13.97±0.00 | 35.16±0.00 |
| EMKMC | 31.46±0.00 | 8.34±0.00 | 32.76±0.00 | 31.05±0.00 | 6.77±0.00 | 31.34±0.00 | 28.53±0.00 | 6.78±0.00 | 29.21±0.00 |
| FastMICE | 35.05±0.00 | 12.78±0.00 | 33.27±0.00 | 34.05±0.00 | 12.18±0.00 | 32.16±0.00 | 33.14±0.00 | 11.66±0.00 | 31.37±0.00 |
| FDAGF | 48.65±3.61 | 25.65±5.05 | 49.18±2.98 | 46.38±2.41 | 25.15±4.64 | 48.71±2.28 | 43.04±3.52 | 22.12±3.17 | 42.68±0.94 |
| MVSC-HFD | 39.06±1.06 | 13.37±0.83 | 39.27±0.87 | 34.89±2.78 | 9.77±1.43 | 35.47±2.91 | 32.89±2.55 | 8.26±0.77 | 33.71±2.17 |
| **Ours** | **50.57±0.05** | **26.43±0.09** | **50.83±0.04** | **48.52±0.05** | **25.22±0.13** | **49.06±0.05** | **46.80±0.01** | **22.36±0.01** | **47.32±0.00** |

- Non-anchor-based clustering methods such as HCP-IMSC and SCSL, fail to operate correctly on slightly larger datasets like MNIST. In contrast, the proposed MCAHC can function reliably in large-scale missing scenarios while still achieving satisfactory results, which demonstrates MCAHC's relatively stronger practicality.

## 5.3 TIME COMPARISON

We present the runtime results of various comparison methods and MCAHC across different datasets, as shown in Figure 2. It should be noted that the vertical axis of the figure employs a logarithmic scale to represent runtime, enabling a more intuitive comparison of the efficiency among different methods. The figure reveals that MCAHC achieves shorter runtime than most comparison methods across the majority of multi-view datasets. For cases where results could not be obtained due to insufficient memory, the corresponding histogram column in the figure remains blank. Thus, MCAHC not only delivers superior clustering results on diverse datasets but also maintains high computational efficiency.

## 5.4 ABLATION

To evaluate the contribution of the hypergraph (HG) module, we compared two variants: w/o HG, which removes HG and retains only the point–anchor bipartite graph; and HG, our proposed anchor-guided hypergraph that models higher-order sample relationships by forming hyperedges around shared anchors. Experiments across diverse datasets and varying proportions of missing views demonstrate that HG consistently outperforms the baseline methods on ACC/NMI/PUR metrics. These results indicate that hyperedges, by jointly make samples connected to the same anchor point, better preserve clustering structures while suppressing cross-view imbalance and noise, thereby achieving more robust and generalizable clustering (see Table 4).

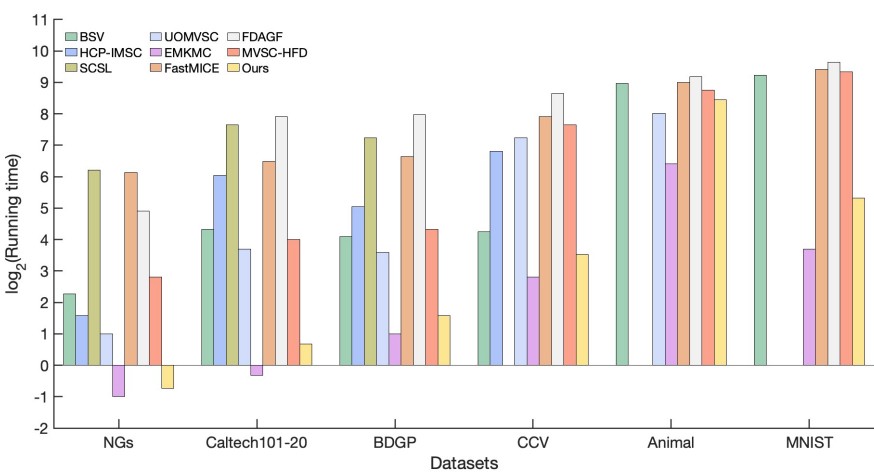

Figure 2: Running time on different datasets.

Table 3: Clustering Results on Datasets

| Method | CCV | | | | | | | | |
|---|---|---|---|---|---|---|---|---|---|
| | 30% | | | 50% | | | 70% | | |
| | ACC | NMI | PUR | ACC | NMI | PUR | ACC | NMI | PUR |
| BSV | 19.26±0.06 | 9.86±0.05 | 17.40±0.08 | 18.35±0.05 | 13.90±0.03 | 20.56±0.04 | 17.38±0.04 | 12.88±0.02 | 19.35±0.03 |
| HCP-IMSC | 10.78±0.07 | 10.76±0.11 | 10.67±0.09 | 10.03±0.09 | 9.91±0.15 | 10.15±0.05 | 9.42±0.07 | 9.13±0.07 | 9.95±0.15 |
| SCSL | N/A | N/A | N/A | N/A | N/A | N/A | N/A | N/A | N/A |
| UOMVSC | 10.91±0.01 | 10.89±0.04 | 10.80±0.04 | 10.21±0.00 | 10.86±0.01 | 9.85±0.00 | 9.45±0.01 | 9.79±0.02 | 9.21±0.02 |
| EMKMC | 11.25±0.00 | 5.77±0.00 | 15.46±0.00 | 10.84±0.00 | 5.73±0.00 | 15.23±0.00 | 10.44±0.00 | 2.98±0.00 | 14.35±0.00 |
| FastMICE | 20.12±0.23 | 8.23±0.09 | 21.37±0.08 | 19.23±0.15 | 7.34±0.07 | 20.41±0.07 | 15.23±0.03 | 4.25±0.17 | 18.23±0.07 |
| FDAGF | 10.50±1.03 | 5.97±0.03 | 19.80±3.65 | 10.12±3.41 | 5.56±0.05 | 19.18±3.36 | 9.45±2.15 | 5.16±0.06 | 18.72±2.78 |
| MVSC-HFD | 20.58±0.00 | 14.41±0.00 | 23.40±0.00 | 18.74±0.00 | 12.89±0.00 | 21.79±0.00 | 16.67±0.00 | 11.13±0.04 | 19.43±0.00 |
| **Ours** | **23.25±0.17** | **16.58±0.04** | **25.96±0.05** | **21.63±0.05** | **15.48±0.04** | **25.09±0.05** | **18.25±0.05** | **13.82±0.07** | **22.02±0.08** |

| Method | Animal | | | | | | | | |
|---|---|---|---|---|---|---|---|---|---|
| | 30% | | | 50% | | | 70% | | |
| | ACC | NMI | PUR | ACC | NMI | PUR | ACC | NMI | PUR |
| BSV | 15.32±0.08 | 10.11±0.04 | 16.38±0.12 | 14.69±0.07 | 9.28±0.05 | 15.55±0.03 | 13.58±0.09 | 7.95±0.09 | 14.68±0.03 |
| HCP-IMSC | N/A | N/A | N/A | N/A | N/A | N/A | N/A | N/A | N/A |
| SCSL | N/A | N/A | N/A | N/A | N/A | N/A | N/A | N/A | N/A |
| UOMVSC | 13.45±1.79 | 11.56±3.64 | 17.78±1.36 | 12.13±2.13 | 10.45±2.79 | 16.42±1.63 | 10.67±2.31 | 9.37±1.56 | 14.41±3.14 |
| EMKMC | 11.45±0.00 | 6.88±0.00 | 11.32±0.00 | 10.43±0.00 | 6.82±0.00 | 10.29±0.00 | 11.14±0.00 | 5.85±0.00 | 11.28±0.00 |
| FastMICE | 9.08±0.00 | 8.18±0.00 | 11.23±0.00 | 9.18±0.00 | 8.17±0.00 | 11.34±0.00 | 8.87±0.00 | 7.58±0.00 | 10.93±0.00 |
| FDAGF | 15.71±0.14 | 9.15±0.25 | 16.55±0.47 | 13.67±0.31 | 7.11±0.14 | 15.11±0.25 | 12.52±0.65 | 6.44±0.23 | 14.51±0.26 |
| MVSC-HFD | **17.60±0.22** | **13.19±0.18** | 20.52±0.10 | 16.13±0.25 | **11.85±0.20** | 19.59±0.13 | 16.15±0.45 | 11.09±0.46 | 19.22±0.57 |
| **Ours** | 17.30±0.00 | 12.65±0.18 | **20.56±0.19** | **16.79±0.07** | 11.59±0.02 | **19.85±0.04** | **16.46±0.01** | **11.21±0.01** | **19.38±0.02** |

| Method | MNIST | | | | | | | | |
|---|---|---|---|---|---|---|---|---|---|
| | 30% | | | 50% | | | 70% | | |
| | ACC | NMI | PUR | ACC | NMI | PUR | ACC | NMI | PUR |
| BSV | 75.88±0.49 | 74.67±0.19 | 78.58±0.33 | 66.49±0.69 | 65.39±0.27 | 69.45±0.49 | 60.69±0.52 | 58.93±0.22 | 62.66±0.38 |
| HCP-IMSC | N/A | N/A | N/A | N/A | N/A | N/A | N/A | N/A | N/A |
| SCSL | N/A | N/A | N/A | N/A | N/A | N/A | N/A | N/A | N/A |
| UOMVSC | N/A | N/A | N/A | N/A | N/A | N/A | N/A | N/A | N/A |
| EMKMC | 71.21±0.30 | 70.88±0.22 | 72.32±0.43 | 70.43±0.23 | 70.82±0.43 | 71.29±0.20 | 70.14±0.00 | 70.25±0.25 | 71.18±0.30 |
| FastMICE | 97.45±0.00 | 96.08±0.00 | 97.89±0.00 | 97.53±0.01 | 95.52±0.00 | 97.68±0.00 | 97.24±0.01 | 95.05±0.00 | 96.89±0.01 |
| FDAGF | 98.64±0.15 | 96.12±0.47 | 97.08±0.95 | 98.23±0.21 | 95.89±0.11 | 97.10±0.23 | 98.05±0.36 | 95.01±0.24 | 96.85±0.34 |
| MVSC-HFD | 75.88±4.86 | 74.67±1.92 | 78.58±3.34 | 66.50±6.99 | 65.39±2.77 | 69.45±4.88 | 60.69±5.21 | 58.93±2.19 | 62.66±3.76 |
| **Ours** | **98.70±0.00** | **96.27±0.00** | **98.42±0.00** | **98.59±0.00** | **96.59±0.00** | **98.42±0.00** | **98.36±0.00** | **95.16±0.00** | **98.32±0.00** |

## 5.5 CONVERGENCE

During the iteration process, we plotted the variation curve of the objective function value. As shown in Figure 3, the objective function value monotonically decreases with increasing iteration count, typically converging after several iterations. Furthermore, we observed that the algorithm exhibits rapid convergence properties, usually reaching a stable state within 20 iterations. These findings undoubtedly validate the convergence of MCAHC.

Table 4: Hypergraph Ablation Results

| AB | MR | NGs | | | Caltech101-20 | | | BDGP | | | CCV | | | Animal | | | MNIST | | |
|---|---|---|---|---|---|---|---|---|---|---|---|---|---|---|---|---|---|---|---|
| | | ACC | NMI | PUR | ACC | NMI | PUR | ACC | NMI | PUR | ACC | NMI | PUR | ACC | NMI | PUR | ACC | NMI | PUR |
| w/o HG | 30% | 90.80 | 77.44 | 90.80 | 51.26 | 53.48 | 71.22 | 48.26 | 24.32 | 48.20 | 21.71 | 16.17 | 24.14 | 15.65 | 11.62 | 18.31 | 98.33 | 95.43 | 97.89 |
| w/ HG | | **94.20** | **83.53** | **94.20** | **55.23** | **58.51** | **76.34** | **50.57** | **26.43** | **50.83** | **23.25** | **16.58** | **25.96** | **17.30** | **12.65** | **20.56** | **98.60** | **95.87** | **98.32** |
| w/o HG | 50% | 89.40 | 73.33 | 89.40 | 52.01 | 50.38 | 69.21 | 44.31 | 22.68 | 45.31 | 18.35 | 13.82 | 21.90 | 14.83 | 10.80 | 17.61 | 98.21 | 95.24 | 97.89 |
| w/ HG | | **91.20** | **77.38** | **91.20** | **55.34** | **56.07** | **73.34** | **48.52** | **25.22** | **49.06** | **21.63** | **15.48** | **25.09** | **16.79** | **11.59** | **19.85** | **98.49** | **95.59** | **98.32** |
| w/o HG | 70% | 83.60 | 62.93 | 83.60 | 47.56 | 50.12 | 70.23 | 42.98 | 18.77 | 43.23 | 18.08 | 13.16 | 21.35 | 15.35 | 10.61 | 16.93 | 98.12 | 94.95 | 97.89 |
| w/ HG | | **87.42** | **69.37** | **87.42** | **51.53** | **53.91** | **73.72** | **46.80** | **21.36** | **47.32** | **18.25** | **13.82** | **22.02** | **16.46** | **11.21** | **19.38** | **98.36** | **95.16** | **98.32** |

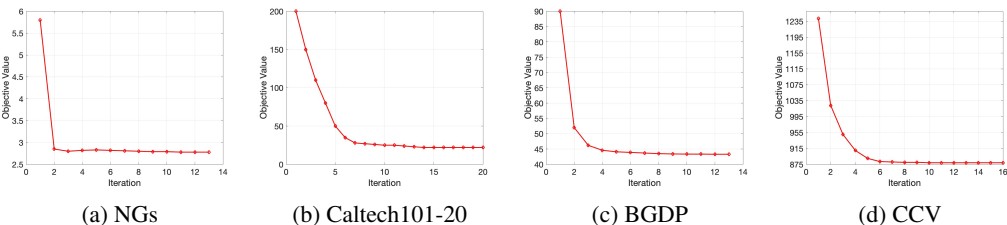

(a) NGs      (b) Caltech101-20      (c) BGDP      (d) CCV

Figure 3: The objective value on on different datasets. (a) NGs (b) Caltech101-20 (c) BDGP (d) CCV.

## 5.6 PARAMETER SENSITIVITY

MCAHC incorporates two hyperparameters, $\lambda_1$ and $\lambda_2$, which govern the hypergraph penalty term and the missingness penalty term respectively. This subsection investigates the influence of these parameters via a grid search method under a 50% missingness rate. Specifically, we set the range for $\lambda_1$ to $10^{-5}, 10^{-2}, \ldots, 10^0$ and the range for $\lambda_2$ to $10^{-4}, 10^{-2}, \ldots, 10^1$. We documented the clustering performance of MCAHC under various parameter combinations, as illustrated in Figure 4. The figure indicates that the optimal values for $\lambda_1$ and $\lambda_2$ lie within the ranges $10^{-5}$ to $10^{-2}$ and $10^{-5}$ to $10^{-1}$, respectively. This phenomenon demonstrates that the MCAHC we propose exhibits stable performance across a wide range of parameters, empirically validating its efficiency and robustness.

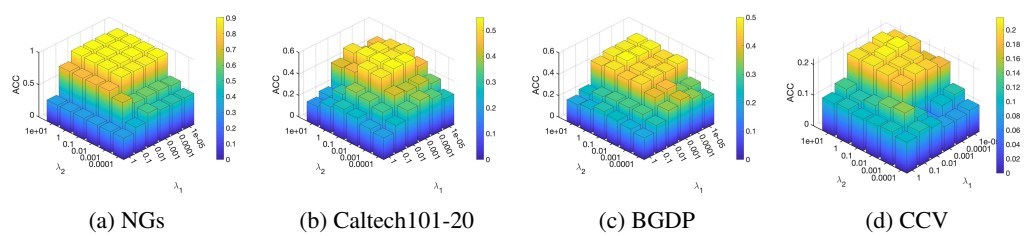

(a) NGs      (b) Caltech101-20      (c) BGDP      (d) CCV

Figure 4: ACC with different parameter combinations across different datasets (a) NGs (b) Caltech101-20 (c) BDGP (d) CCV.

## 6 CONCLUSION

We propose MCAHC, a missing-aware consensus anchor-guided hypergraph framework for incomplete multi-view clustering. By introducing anchor-guided hypergraphs, MCAHC captures high-order anchor-sample group interactions. Simultaneously, it incorporates reweighting mechanisms at both sample and view levels to explicitly mitigate view imbalance and missingness while suppressing the impact of missing data on anchor selection. The hypergraph Laplacian regularization term further enhances cross-view structural consistency without sacrificing scalability driven by anchors. Extensive experiments across multiple datasets and varying missingness rates demonstrate that MCAHC achieves stable performance improvements and exhibits strong robustness against noise.

ETHICS STATEMENT

This paper does not involve any potential ethics issues.

REPRODICIBILITY STATEMENT

We have submitted the code and datasets to facilitate reproduction of our results.

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

APPENDIX

## A  LLM USAGE

We used a large language model for language editing, including spelling and grammar checks.

## B  SYMBOL SUMMARY

$v$: the number of views;
$n$: the number of samples;
$k$: the number of clusters;
$m$: the number of anchors;
$d_p$: the feature dimension on view $p$;
$l$: the dimension of the consensus subspace;
$\mathbf{X}_p$: the data matrix on view $p$, $d_p \times n$;
$\mathbf{W}_p$: the projection matrix on view $p$ to the consensus space, $d_p \times l$;
$\mathbf{A}$: the consensus anchor matrix, $l \times m$;
$\mathbf{Z}$: the anchor graph, $m \times n$ ;
$\mathbf{z}_i$: the $i$-th column of $\mathbf{Z}$, $m \times 1$;
$\beta_p$: the view-weight vector;
$s_p^{(i)}$: indicator that sample $i$ is observed in view $p$ (1) or missing (0);
$\mathbf{S}_p$: the diagonal observation mask on view $p$, $n \times n$;
$q_i$: the missing rate of sample $i$;
$\mathbf{Q}$: the diagonal matrix $\mathrm{Diag}(q_i)$, $n \times n$;
$m_i$: the sample weight, e.g., $m_i = e^{-\gamma q_i}$;
$\mathbf{M}$: the diagonal matrix $\mathrm{Diag}(m_i)$, $n \times n$;
$\gamma$: the decay coefficient controlling the missingness penalty;
$\mathbf{H}$: the hypergraph incidence matrix, $(n+m) \times m$;
$\mathbf{D}_v$, $\mathbf{D}_e$: the vertex-degree and hyperedge-degree diagonal matrices;
$\mathbf{L}_H$: the normalized hypergraph Laplacian, $(n+m) \times (n+m)$;
$\mathbf{Z}_s$: the sample embeddings in the consensus space, $l \times n$;
$\mathbf{Z}_{\mathrm{aug}}$: the joint embeddings of samples and anchors, $l \times (n+m)$;
$\lambda_1$, $\lambda_2$: the coefficients of the two regularizers;

## C  DETAILED DERIVATIONS OF OPTIMIZATION

With constants $\lambda_1, \lambda_2 \geq 0$, the problem is

$$\min_{\{\mathbf{W}_p\}, \mathbf{A}, \mathbf{Z}, \beta} \sum_{p=1}^{v} \beta_p^2 \left\| \left(\mathbf{W}_p^\top \mathbf{X}_p - \mathbf{A}\mathbf{Z}\right) \mathbf{S}_p \mathbf{M}^{\frac{1}{2}} \right\|_F^2 \; + \; \lambda_1 \, \mathrm{Tr}\left(\mathbf{Z}_{\mathrm{aug}} \mathbf{L}_H \mathbf{Z}_{\mathrm{aug}}^\top\right) \; + \; \lambda_2 \left\| \mathbf{Z}\mathbf{Q}^{\frac{1}{2}} \right\|_F^2.$$

1. UPDATE OF $\mathbf{W}_p$

When fixing all other variables unrelated to $\mathbf{W}_p$, the subproblem for updating $\mathbf{W}_p$ is

$$\min_{\mathbf{W}_p^\top \mathbf{W}_p = \mathbf{I}} \left\| \left(\mathbf{W}_p^\top \mathbf{X}_p - \mathbf{A}\mathbf{Z}\right) \mathbf{S}_p \mathbf{M}^{\frac{1}{2}} \right\|_F^2.$$

For a fixed $p$, define the residual $\mathbf{E}_p = (\mathbf{W}_p^\top \mathbf{X}_p - \mathbf{A}\mathbf{Z})\mathbf{S}_p \mathbf{M}^{\frac{1}{2}}$. Using the properties of trace, and $\mathbf{S}_p^2 = \mathbf{S}_p$, $\mathbf{M}^{\frac{1}{2}}\mathbf{M}^{\frac{1}{2}} = \mathbf{M}$,

$$\begin{aligned}
\|\mathbf{E}_p\|_F^2 &= \mathrm{Tr}\Big( (\mathbf{W}_p^\top \mathbf{X}_p \mathbf{S}_p \mathbf{M}^{\frac{1}{2}} - \mathbf{A}\mathbf{Z}\mathbf{S}_p \mathbf{M}^{\frac{1}{2}})(\mathbf{W}_p^\top \mathbf{X}_p \mathbf{S}_p \mathbf{M}^{\frac{1}{2}} - \mathbf{A}\mathbf{Z}\mathbf{S}_p \mathbf{M}^{\frac{1}{2}})^\top \Big) \\
&= \mathrm{Tr}\big(\mathbf{W}_p^\top \mathbf{X}_p \mathbf{S}_p \mathbf{M}\mathbf{X}_p^\top \mathbf{W}_p\big) + \mathrm{Tr}\big(\mathbf{A}\mathbf{Z}\mathbf{S}_p \mathbf{M}\mathbf{Z}^\top \mathbf{A}^\top\big) \\
&\quad - 2\,\mathrm{Tr}\big(\mathbf{W}_p^\top \mathbf{X}_p \mathbf{S}_p \mathbf{M}\mathbf{Z}^\top \mathbf{A}^\top\big).
\end{aligned}$$

For fixed $\mathbf{A}$, $\mathbf{Z}$, the second term is constant in $\mathbf{W}_p$, hence minimizing $\|\mathbf{E}_p\|_F^2$ is equivalent to

$$\max_{\mathbf{W}_p^\top \mathbf{W}_p = \mathbf{I}} \mathrm{Tr}\big(\mathbf{W}_p^\top \mathbf{G}_p\big), \qquad \mathbf{G}_p = \mathbf{X}_p \mathbf{S}_p \mathbf{M} \mathbf{Z}^\top \mathbf{A}^\top.$$

Let the thin SVD be $\mathbf{G}_p = \mathbf{U}_p \boldsymbol{\Sigma}_p \mathbf{V}_p^\top$. The orthogonal Procrustes solution gives

$$\mathbf{W}_p^\star = \mathbf{U}_p \mathbf{V}_p^\top.$$

## 2. UPDATE OF $\mathbf{A}$

Fixing $\{\mathbf{W}_p\}$, $\mathbf{Z}$, and $\beta$, the subproblem for updating $\mathbf{A}$ is

$$\min_{\mathbf{A}^\top \mathbf{A} = \mathbf{I}} \sum_{p=1}^{v} \beta_p^2 \big\|\big(\mathbf{W}_p^\top \mathbf{X}_p - \mathbf{A}\mathbf{Z}\big) \mathbf{S}_p \mathbf{M}^{\frac{1}{2}}\big\|_F^2.$$

From the previous expansion, for each $p$ we have

$$\|\mathbf{E}_p\|_F^2 = \mathrm{Tr}\big(\mathbf{W}_p^\top \mathbf{X}_p \mathbf{S}_p \mathbf{M} \mathbf{X}_p^\top \mathbf{W}_p\big) + \mathrm{Tr}\big(\mathbf{A}\mathbf{Z}\mathbf{S}_p \mathbf{M} \mathbf{Z}^\top \mathbf{A}^\top\big) - 2\,\mathrm{Tr}\big(\mathbf{W}_p^\top \mathbf{X}_p \mathbf{S}_p \mathbf{M} \mathbf{Z}^\top \mathbf{A}^\top\big).$$

For the second term, using the cyclic property of the trace and the orthogonality constraint $\mathbf{A}^\top \mathbf{A} = \mathbf{I}$, we obtain

$$\mathrm{Tr}\big(\mathbf{A}\mathbf{Z}\mathbf{S}_p \mathbf{M} \mathbf{Z}^\top \mathbf{A}^\top\big) = \mathrm{Tr}\big(\mathbf{A}^\top \mathbf{A}\,\mathbf{Z}\mathbf{S}_p \mathbf{M}\mathbf{Z}^\top\big) = \mathrm{Tr}\big(\mathbf{Z}\mathbf{S}_p \mathbf{M}\mathbf{Z}^\top\big),$$

which is constant with respect to $\mathbf{A}$. Therefore, discarding the terms independent of $\mathbf{A}$, the optimization reduces to

$$\sum_{p=1}^{v} \beta_p^2 \|\mathbf{E}_p\|_F^2 = -2\,\mathrm{Tr}\Big(\mathbf{A}^\top \sum_{p=1}^{v} \beta_p^2\,\mathbf{W}_p^\top \mathbf{X}_p \mathbf{S}_p \mathbf{M}\mathbf{Z}^\top\Big) + \text{const}.$$

Hence, updating $\mathbf{A}$ is equivalent to solving

$$\max_{\mathbf{A}^\top \mathbf{A} = \mathbf{I}} \mathrm{Tr}\big(\mathbf{A}^\top \mathbf{P}\big), \qquad \mathbf{P} = \sum_{p=1}^{v} \beta_p^2\,\mathbf{W}_p^\top \mathbf{X}_p \mathbf{S}_p \mathbf{M}\mathbf{Z}^\top.$$

Although the augmented representation includes $\mathbf{A}$, the Laplacian quadratic form $\lambda_1 \mathrm{Tr}\big(\mathbf{Z}_{\mathrm{aug}} \mathbf{L}_H \mathbf{Z}_{\mathrm{aug}}^\top\big)$ is in fact invariant with respect to $\mathbf{A}$. To see this, partition the hypergraph Laplacian over samples and anchors:

$$\mathbf{L}_H = \begin{bmatrix} \mathbf{L}_{dd} & \mathbf{L}_{da} \\ \mathbf{L}_{ad} & \mathbf{L}_{aa} \end{bmatrix}, \qquad \mathbf{Z}_{\mathrm{aug}} = [\, \mathbf{A}\mathbf{Z},\ \mathbf{A}\,].$$

Expanding the trace gives

$$\mathrm{Tr}\big(\mathbf{Z}_{\mathrm{aug}} \mathbf{L}_H \mathbf{Z}_{\mathrm{aug}}^\top\big) = \mathrm{Tr}\big(\mathbf{A}\mathbf{Z}\mathbf{L}_{dd}\mathbf{Z}^\top \mathbf{A}^\top\big) + 2\,\mathrm{Tr}\big(\mathbf{A}\mathbf{Z}\mathbf{L}_{da}\mathbf{A}^\top\big) + \mathrm{Tr}\big(\mathbf{A}\mathbf{L}_{aa}\mathbf{A}^\top\big).$$

Because $\mathbf{A}$ is column-orthonormal ($\mathbf{A}^\top \mathbf{A} = \mathbf{I}$), applying this to every term above yields that $\mathrm{Tr}\big(\mathbf{Z}_{\mathrm{aug}} \mathbf{L}_H \mathbf{Z}_{\mathrm{aug}}^\top\big)$ contains no $\mathbf{A}$. Consequently, $\frac{\partial}{\partial \mathbf{A}} \mathrm{Tr}\big(\mathbf{Z}_{\mathrm{aug}} \mathbf{L}_H \mathbf{Z}_{\mathrm{aug}}^\top\big) = \mathbf{0}$. By the same reasoning, the term $\lambda_2 \|\mathbf{Z}\mathbf{Q}^{1/2}\|_F^2$ is independent of $\mathbf{A}$ as it does not involve $\mathbf{A}$ at all.

Let $\mathbf{P} = \mathbf{U}\boldsymbol{\Sigma}\mathbf{V}^\top$ be the SVD of $\mathbf{P}$. The optimal solution is then given by $\mathbf{A}^\star = \mathbf{U}\mathbf{V}^\top$.

## 3. UPDATE OF $\mathbf{Z}$

Fixing $\{\mathbf{W}_p\}$, $\mathbf{A}$, and $\beta_p$, the $\mathbf{Z}$-subproblem is a convex quadratic with simplex constraints:

$$\min_{\mathbf{Z} \geq 0,\ \mathbf{Z}^\top \mathbf{1} = \mathbf{1}} \sum_{p=1}^{v} \beta_p^2 \big\|\big(\mathbf{W}_p^\top \mathbf{X}_p - \mathbf{A}\mathbf{Z}\big) \mathbf{S}_p \mathbf{M}^{\frac{1}{2}}\big\|_F^2 + \lambda_1 \mathrm{Tr}\big(\mathbf{Z}_{\mathrm{aug}}^\top \mathbf{L}_H \mathbf{Z}_{\mathrm{aug}}\big) + \lambda_2 \|\mathbf{Z}\mathbf{Q}^{\frac{1}{2}}\|_F^2.$$

Collecting the $\mathbf{Z}$-dependent part of the data term via the same expansion gives

$$\sum_{p=1}^{v} \beta_p^2 \|\mathbf{E}_p\|_F^2 = \mathrm{Tr}(\mathbf{Z}\,\mathbf{C}\,\mathbf{Z}^\top) - 2\,\mathrm{Tr}(\mathbf{Z}^\top \mathbf{A}^\top \mathbf{B}) + \text{const}, \quad \mathbf{C} = \sum_{p=1}^{v} \beta_p^2 \mathbf{S}_p \mathbf{M}, \;\; \mathbf{B} = \sum_{p=1}^{v} \beta_p^2 \mathbf{W}_p^\top \mathbf{X}_p \mathbf{S}_p \mathbf{M},$$

where we used $\mathrm{Tr}(\mathbf{A}\mathbf{Z}\mathbf{S}_p \mathbf{M}\mathbf{Z}^\top \mathbf{A}^\top) = \mathrm{Tr}(\mathbf{Z}\mathbf{S}_p \mathbf{M}\mathbf{Z}^\top)$ (from $\mathbf{A}^\top \mathbf{A} = \mathbf{I}$) and $-2\sum_p \beta_p^2 \mathrm{Tr}(\mathbf{W}_p^\top \mathbf{X}_p \mathbf{S}_p \mathbf{M}\mathbf{Z}^\top \mathbf{A}^\top) = -2\,\mathrm{Tr}(\mathbf{Z}^\top \mathbf{A}^\top \mathbf{B})$. For the Laplacian term, with $\mathbf{Z}_{\text{aug}} = [\,\mathbf{A}\mathbf{Z},\ \mathbf{A}\,]$ and $\mathbf{L}_H = \begin{bmatrix} \mathbf{L}_{dd} & \mathbf{L}_{da} \\ \mathbf{L}_{ad} & \mathbf{L}_{aa} \end{bmatrix}$, we obtain the blockwise trace expansion

$$\mathrm{Tr}(\mathbf{Z}_{\text{aug}} \mathbf{L}_H \mathbf{Z}_{\text{aug}}^\top) = \mathrm{Tr}(\mathbf{A}\mathbf{Z}\mathbf{L}_{dd}\mathbf{Z}^\top \mathbf{A}^\top) + 2\,\mathrm{Tr}(\mathbf{A}\mathbf{Z}\mathbf{L}_{da}\mathbf{A}^\top) + \mathrm{Tr}(\mathbf{A}\mathbf{L}_{aa}\mathbf{A}^\top).$$

so the $\mathbf{Z}$-dependent contribution is $\lambda_1\big(\mathrm{Tr}(\mathbf{Z}\mathbf{L}_{dd}\mathbf{Z}^\top) + 2\,\mathrm{Tr}(\mathbf{Z}\mathbf{L}_{da})\big)$. The missing-rate regularizer satisfies $\|\mathbf{Z}\mathbf{Q}^{\frac{1}{2}}\|_F^2 = \mathrm{Tr}(\mathbf{Z}\mathbf{Q}\mathbf{Z}^\top)$. Putting pieces together and discarding constants yields

$$f(\mathbf{Z}) = \mathrm{Tr}(\mathbf{Z}\mathbf{K}\mathbf{Z}^\top) - 2\,\mathrm{Tr}(\mathbf{Z}^\top \mathbf{J}), \qquad \mathbf{K} = \mathbf{C} + \lambda_1 \mathbf{L}_{dd} + \lambda_2 \mathbf{Q}, \quad \mathbf{J} = \mathbf{A}^\top \mathbf{B} - \lambda_1 \mathbf{L}_{ad},$$

where $\mathbf{K} \succeq \mathbf{0}$ since $\mathbf{C}, \mathbf{L}_{dd}, \mathbf{Q} \succeq \mathbf{0}$. The gradient and a global Lipschitz constant are

$$\nabla f(\mathbf{Z}) = 2\mathbf{Z}\mathbf{K} - 2\mathbf{J}, \qquad L \geq 2\lambda_{\max}(\mathbf{K}).$$

A projected FISTA scheme proceeds as follows: initialize $t_0 = 1$ and $\mathbf{Z}^{(0)} = \mathbf{Z}^{(-1)}$, then for $k = 0, 1, 2, \dots$

$$t_{k+1} = \frac{1 + \sqrt{1 + 4t_k^2}}{2}, \mathbf{Y}^{(k)} = \mathbf{Z}^{(k)} + \frac{t_k - 1}{t_{k+1}}\big(\mathbf{Z}^{(k)} - \mathbf{Z}^{(k-1)}\big), \widetilde{\mathbf{Z}} = \mathbf{Y}^{(k)} - \frac{1}{L}\big(2\mathbf{Y}^{(k)}\mathbf{K} - 2\mathbf{J}\big),$$

$$\mathbf{Z}_{:,j}^{(k+1)} = \Pi_\Delta\big(\widetilde{\mathbf{Z}}_{:,j}\big) \ \text{for} \ j = 1, \dots, n, \ \ \Delta = \{z \in \mathbb{R}^m \mid z \geq 0, \ \mathbf{1}^\top z = 1\},$$

where the Euclidean projection $\Pi_\Delta$ onto the probability simplex is computed columnwise by sorting: for $u = \widetilde{\mathbf{Z}}_{:,j}$, let $\mu$ be $u$ sorted in descending order, find $\rho = \max\{k : \mu_k + \frac{1}{k}\big(1 - \sum_{i=1}^{k} \mu_i\big) > 0\}$ and set $\theta = \frac{1}{\rho}\big(\sum_{i=1}^{\rho} \mu_i - 1\big)$, then $(\Pi_\Delta(u))_i = \max\{u_i - \theta, 0\}$.

## 4. UPDATE OF $\beta$

Fixing $\{\mathbf{W}_p\}$, $\mathbf{A}$, and $\mathbf{Z}$, the $\beta$-subproblem is

$$\min_{\beta \geq 0, \ \mathbf{1}^\top \beta = 1} \sum_{p=1}^{v} \beta_p^2 R_p^2, \qquad R_p = \big\|(\mathbf{W}_p^\top \mathbf{X}_p - \mathbf{A}\mathbf{Z})\mathbf{S}_p \mathbf{M}^{\frac{1}{2}}\big\|_F.$$

Introduce the Lagrangian $\mathcal{L}(\beta, \lambda, \mu) = \sum_{p=1}^{v} \beta_p^2 R_p^2 - \lambda\big(\sum_{p=1}^{v} \beta_p - 1\big) - \sum_{p=1}^{v} \mu_p \beta_p$ with multipliers $\lambda \in \mathbb{R}$ and $\mu_p \geq 0$.

For any $p$ with $R_p > 0$, optimality yields $\beta_p > 0$ and hence $\mu_p = 0$, giving $2\beta_p R_p^2 - \lambda = 0 \Rightarrow \beta_p = \frac{\lambda}{2R_p^2}$. Enforcing $\sum_p \beta_p = 1$ gives $\lambda = \frac{2}{\sum_{u=1}^{v} R_u^{-2}}$, and therefore

$$\beta_p^\star = \frac{R_p^{-2}}{\sum_{u=1}^{v} R_u^{-2}}, \qquad p = 1, \dots, v.$$

## 5. UPDATE OF $\mathbf{L}_H$

Given the current $\mathbf{Z}$:

1. For each sample $i$, take indices of its $T$ most similar anchors by the largest entries of $z_{j,i}$; create a hyperedge for each selected anchor $j$ that connects the data vertices. Form the incidence matrix $\mathbf{H} \in \mathbb{R}^{(n+m)\times E}$ by $\mathbf{H}_{i,e} = 1$ if vertex $i$ participates in hyperedge $e$.

2. Optionally add an anchor self-edge for each anchor vertex to stabilize degrees.

3. Compute $\mathbf{D}_v = \mathrm{Diag}(\mathbf{H1})$, $\mathbf{D}_e = \mathrm{Diag}(\mathbf{1}^\top \mathbf{H})$, and

$$\mathbf{L}_H = \mathbf{I} - \mathbf{D}_v^{-\frac{1}{2}} \mathbf{H} \mathbf{D}_e^{-1} \mathbf{H}^\top \mathbf{D}_v^{-\frac{1}{2}}.$$

