# OpenReview forum: "A Consensus Anchor-Guided Hypergraph Framework for Incomplete Multi-View Clustering"
_ICLR.cc/2026/Conference — ICLR 2026 Conference Withdrawn Submission_

### Official Review · Reviewer_u4zk · 2025-10-27

**Soundness:** 2
**Presentation:** 2
**Contribution:** 2
**Rating:** 4
**Confidence:** 4

**Summary:**

This paper proposes an incomplete multi-view clustering framework named MCAHC. It captures the high-order relationships among samples by introducing a consensus anchor-guided hypergraph structure and designs sample-level and view-level reweighting mechanisms to address the view missing problem. Combining hypergraph Laplacian regularization and a missing-aware mechanism, this method achieves efficient and robust clustering performance on multiple datasets and in scenarios with high missing rates.

**Strengths:**

1. Introduction of Hypergraph Structure: It extends the traditional binary anchor-sample relationship to a high-order relationship, enhancing the model’s ability to capture complex structures.

2. Missing-Aware Mechanism: Through sample-level and view-level reweighting, it effectively mitigates the impact of view imbalance and missing data on clustering.

3. Efficient Optimization Algorithm: It adopts an alternating optimization strategy, featuring good convergence and computational efficiency, and is suitable for large-scale data.

4. Strong Robustness and Generalization: It performs excellently on multiple datasets and under high missing rates (up to 70%), outperforming existing baseline methods.

**Weaknesses:**

1. Hyperparameter Sensitivity: The model contains multiple hyperparameters (e.g., $\lambda_1$, $\lambda_2$, $T$), which require careful tuning, and no adaptive selection strategy is provided.

2. Dependence on Anchor Quality: The generation and selection of anchors significantly affect the results, yet the paper does not fully discuss their robustness to missing data.

3. Insufficient Innovation in the proposed objective function: It merely incorporates a regularization term into a common framework, lacking sufficient innovation.

4. Weak Interpretability: Its core components are weak in terms of intuitive understanding and interpretability. It is difficult to clearly explain to domain experts why a sample is assigned to a specific cluster and what kind of semantic group a hyperedge specifically represents.

**Questions:**

1. How to adaptively select hyperparameters to avoid reliance on grid search?

2. Can the hypergraph structure be further integrated with semantic information or prior knowledge to enhance the interpretability of clustering?

---

> ### Author Response · Authors · 2025-11-26
>
> We thank reviewer u4zk for the constructive comments and guidance provided during the revision of this manuscript. We have carefully addressed all concerns and sincerely hope that all issues have been resolved.
>
> **Q1.** Hyperparameter Sensitivity: The model contains multiple hyperparameters (e.g.,
> $\lambda_1$, $\lambda_2$, $T$), which require careful tuning, and no adaptive selection strategy is provided.
>
> **A1.** We would like to clarify that designing an automatic, data-driven tuning mechanism for $\lambda_1, \lambda_2$ and $T$ in incomplete multi-view clustering is itself a non-trivial research problem and goes beyond the scope of this work. To the best of our knowledge, the vast majority of existing IMVC or MVC methods, including those we compare against, also rely on a small grid search rather than an explicit automatic scheme.
>
> In the revision, instead of proposing a heuristic auto-tuning rule, we strengthen the empirical evidence by adding systematic sensitivity analyses foreach key hyperparameter. Concretely, we perform coarse grid searches for $\lambda_1$ and $\lambda_2$, vary $T \in \{2,4,6,8,10\}$ under different missing rates, and search the number of anchors $m \in \{k,2k,3k,4k,5k\}$. The results show that MCAHC is robust within a wide range of settings and does not rely on delicate hyperparameter tuning, even though a fully automatic selection rule is not provided.
>
>
> **Q2.** Dependence on Anchor Quality: The generation and selection of anchors significantly affect the results, yet the paper does not fully discuss their robustness to missing data.
>
> **A2.** We agree that anchor quality significantly impacts anchor-based multi-view clustering. In the revised version, we will supplement more comprehensive explanations to demonstrate the model's robustness under missing data. First, our MCAHC framework automatically mitigates the impact of highly missing samples on anchor learning through the missing-aware weight matrices $M$ and $Q$, preventing anchors from being dominated by noisy or severely missing samples. Second, anchor updates employ multi-view consistency constraints combined with an orthogonal Procrustes closed-form solution. This ensures anchors are jointly calibrated by all observable views in each iteration, thereby preventing bias from single or incomplete views.
>
> **Q3.** Insufficient Innovation in the proposed objective function: It merely incorporates a regularization term into a common framework, lacking sufficient innovation.
>
> **A3.** We recognize the importance of distinguishing our proposed method from existing frameworks, and will therefore further emphasize the core innovations of the objective function in the revised manuscript. While the overall structure remains grounded in minimizing reconstruction error, our objective function is not a simple aggregation of conventional regularization terms. Instead, it is systematically designed around an “anchor-guided missing-perception hypergraph structure,” manifested in the following three aspects:
>
> 	1.Anchor-guided hypergraphs represent a novel high-order structural modeling approach. Existing IMVC approaches predominantly employ KNN or sample space composition. Our proposed objective function, however, explicitly incorporates ‘‘anchor-induced higher-order relationships’’ into the optimization target for the first time. Each anchor corresponds to a hyperedge, enabling simultaneous learning of anchors, hyperedges, and consistent embedding structures during optimization. This represents a structural innovation in higher-order modeling, distinct from traditional second-order graph regularization.
>
> 	2. The missing-aware regularization terms M and Q are not part of a conventional framework; instead, they are dynamically constructed based on the actual missing patterns.These terms are not fixed prior regularizers. Rather, they adaptively adjust the strength of anchor learning and hypergraph smoothing according to each sample’s true missing rate, allowing the model to continuously accommodate different missing patterns and noise structures during optimization. This adaptive mechanism has not appeared in existing anchor-based IMVC methods.
>
> 	3.The structural consistency term interacts with the anchor hypergraph, not functioning as an isolated regularization term.
> The hypergraph Laplacian represents higher-order structures generated by anchors. Its influence on cluster structures is achieved not through conventional Laplacian regularization, but by implementing“cross-view unified higher-order structural constraints.”This constitutes a framework-level innovation, not merely the addition of a Laplacian term.
>
> We will incorporate clearer explanations in the revised manuscript, emphasizing that the objective function forms a complete new framework through the synergistic design of “anchor points—hypergraph—missing perception,” rather than simply adding a regularization term to a generic model.

---

> ### Author Response · Authors · 2025-11-26
>
> **Q4.** Weak Interpretability: Its core components are weak in terms of intuitive understanding and interpretability. It is difficult to clearly explain to domain experts why a sample is assigned to a specific cluster and what kind of semantic group a hyperedge specifically represents.
>
> **A4.** Thank you for highlighting this important point. We enhance interpretability by adding intuitive explanations:
>
> 	1.Semantic meaning of hyperedges: Under anchor guidance, a hyperedge represents “the concept center represented by an anchor” and “its highly correlated sample subset,” analogous to high-order consistent groups across views.
>
> 	2.Structural consistency explanation: By analyzing the Laplacian term, we show that the model tends to cluster shared structures—patterns stable across views—into cohesive groups.

---

### Official Review · Reviewer_Jypc · 2025-10-28

**Soundness:** 3
**Presentation:** 2
**Contribution:** 3
**Rating:** 4
**Confidence:** 4

**Summary:**

This paper addresses the Incomplete Multi-View Clustering (IMVC) problem by proposing a novel framework named MCAHC. The core innovation of the method lies in extending the traditional anchor-based bipartite graph model into an anchor-guided hypergraph model, which effectively captures higher-order relationships among samples and anchors. The paper provides a detailed formulation of the objective function, derivations of the optimization algorithm (an alternating optimization procedure involving SVD and FISTA), and a complete algorithmic workflow. The experimental section, validated across multiple datasets and varying missing rates, demonstrates that MCAHC outperforms a range of baseline methods in both clustering performance and computational efficiency. Ablation studies, convergence analysis, and parameter sensitivity analysis further substantiate the effectiveness of its individual components and the overall robustness of the model.

**Strengths:**

1.The integration of a hypergraph into the anchor-guided IMVC framework, combined with sample and view level reweighting mechanisms, represents a clear and significant innovation. This approach effectively mitigates view imbalance and missingness while suppressing the influence of incomplete data on anchor selection. It enhances cross-view structural consistency, and its efficacy is convincingly validated through experiments.

2.The paper provides a very complete and systematic description of the proposed method. This includes the motivation, model formulation, detailed optimization derivations, and algorithm , which together form a systematic and cohesive whole. This thoroughness significantly enhances the paper's reproducibility and represents a valuable contribution to the community.

**Weaknesses:**

1.The paper details the iterative optimization process for the core objective function but provides little discussion on the initialization strategy for the model parameters (e.g., the consensus anchor matrix A). The quality of initializations can influence the optimization trajectory and final results. It would strengthen the paper to include a brief discussion on this aspect in a revised version.

2.The experimental section includes a dedicated discussion on the impact of hyperparameters λ₁ and λ₂. However, it lacks a detailed analysis on the selection or sensitivity of T (the number of anchors each sample connects to in the hypergraph). This parameter crucially influences the hypergraph's topology. The paper would benefit from discussing how T was set and its potential impact on performance.

 3.On Page 1, the phrase "an contrastive learning framework" contains a grammatical error.

**Questions:**

See Weaknesses.

---

> ### Author Response · Authors · 2025-11-26
>
> We extend our sincere gratitude to reviewer jypc for their constructive feedback and guidance on revising this manuscript. We have provided detailed responses to all concerns and sincerely hope that all issues have been adequately addressed.
>
> **Q1.** The paper details the iterative optimization process for the core objective function but provides little discussion on the initialization strategy for the model parameters (e.g., the consensus anchor matrix $A$). The quality of initializations can influence the optimization trajectory and final results. It would strengthen the paper to include a brief discussion on this aspect in a revised version.
>
> **A1.** During the initialization phase, we first compute the missingness level of each sample based on the view indicator matrix, and use this to construct the missing-aware weight matrices $M$ and $Q$. The anchor graph $Z$ is initialized as a truncated identity matrix, ensuring each anchor corresponds to one sample at the outset, thereby providing a starting point for anchor structure learning. The projection matrix $W_p$ and consensus anchor matrix $A$ are not directly set to fixed values. Instead, they are automatically obtained as nonzero matrices satisfying orthogonality constraints during the first update based on the data $X_p$ and the initial anchor assignments. Based on the initial $A$ and $Z$, we further construct an initial anchor-guided hypergraph and its hyper-Laplacian matrix. This initialization method is simple and stable, exhibiting good convergence properties during subsequent alternating updates.
>
> **Q2.** The experimental section includes a dedicated discussion on the impact of hyperparameters $\lambda_1$ and $\lambda_2$. However, it lacks a detailed analysis on the selection or sensitivity of $T$ (the number of anchors each sample connects to in the hypergraph). This parameter crucially influences the hypergraph's topology. The paper would benefit from discussing how $T$ was set and its potential impact on performance.
>
> **A2.** We appreciate the reviewers' comments regarding the hyperparameter $T$, which represents the number of anchors connected to each sample in the hypergraph. We conducted a more detailed sensitivity analysis across two representative datasets (NGs and CCV) at different missing rates.
>
> Specifically, we varied $T \in \{2,4,6,8,10\}$ and report accuracy (ACC), normal matching index (NMI), and purity (PUR) for the NGs and CCV datasets at 30%, 50%, and 70% missingness rates (see Tables 1). The results demonstrate exceptional robustness of MCAHC to the choice of $T$: moderate hyperedge sizes (around $T=6$) achieved slightly superior performance across most settings. This aligns with intuitive expectations: connecting each sample to a small number of ‘‘representative’’ anchors sufficiently aggregates information while avoiding computational redundancy and over-smoothing from excessively large $T$; conversely, too small $T$ may lead to insufficient information. Based on these observations, we set $T=6$ as the default trade-off value for sparsity and information aggregation. This confirms that our anchor-guided hypergraph model does not rely on fine-tuning $T$ even under high missingness conditions.
>
> ### Table 1. Sensitivity of T on NGs and CCV (ACC / NMI / PUR)
> | Dataset | Missing rate | T = 2      | T = 4        | T = 6    | T = 8                          | T = 10         |
> |---------|---------|---------------------------|---------------------------|---------------------------|-------------------------------|---------------------------|
> | NGs     | 30%     | 92.80 / 80.25 / 92.80     | 93.80 / 82.61 / 93.80     | 93.80 / 82.60 / 93.80     | **94.20 / 83.47 / 94.20**     | 93.80 / 82.69 / 93.80     |
> | NGs     | 50%     | 94.20 / 74.80 / 94.20     | 94.20 / 76.94 / 94.20     | **94.80 / 77.26 / 94.80** | 94.80 / 76.03 / 94.80         | 94.00 / 75.16 / 94.00     |
> | NGs     | 70%     | 85.80 / 67.14 / 85.80     | 86.00 / 67.36 / 86.00     | **87.40 / 68.84 / 87.40** | 86.00 / 67.56 / 86.00         | 85.80 / 67.19 / 85.80     |
> | CCV     | 30%     | 23.36 / 16.69 / 26.12     | 23.37 / 16.66 / 26.09     | **23.40 / 16.66 / 26.17** | 23.37 / 16.62 / 26.13         | 23.24 / 16.61 / 25.98     |
> | CCV     | 50%     | 21.73 / 15.40 / 25.14     | 21.62 / 15.46 / 25.06     | 21.57 / 15.40 / 25.06     | **21.68 / 15.51 / 25.18**     | 21.66 / 15.49 / 25.12     |
> | CCV     | 70%     | 18.25 / 13.67 / 22.09     | 18.16 / 13.89 / 22.08     | **18.27 / 13.82 / 22.14** | 18.23 / 13.75 / 22.01         | 18.14 / 13.47 / 21.95     |
>
>
> **Q3.** On Page 1, the phrase "an contrastive learning framework" contains a grammatical error.
>
> **A3.** We thank the reviewer for pointing this out. We will correct the phrase ‘‘contrastive learning framework’’ on page 1 to ensure grammatical accuracy and clarity.

---

### Official Review · Reviewer_k9nC · 2025-10-29

**Soundness:** 3
**Presentation:** 3
**Contribution:** 2
**Rating:** 4
**Confidence:** 4

**Summary:**

This paper aims to address the incomplete MVC task. Traditional methods frequently use anchor learning to avoid large computations, but lack robust learning of anchors and fail to extract high-order correlations. The authors propose MCAHC to overcome these drawbacks and propose an alternating optimization process.

**Strengths:**

1.The code is provided in the appendix, ensuring its reproducibility.

2.The extraction of high-order correlations between anchors and samples is interesting.

3.The experimental results demonstrate the method's superiority.

**Weaknesses:**

1.The novelty appears limited. Anchor learning is widespread in IMVC, and the hypergraph concept has been proposed for IMVC and anchor-based MVC, as seen in [1-2].

2.Most of the compared algorithms are designed for complete MVC. It should be clarified how these methods were extended to the IMVC setting. Furthermore, the paper should justify why more state-of-the-art IMVC methods were not included in the comparison.

3.The authors claim that “These methods lack robust learning of consensus anchors”. However, this claim lacks sufficient theoretical or experimental support.

[1]Chen J, Xu H, Xue J, et al. Incomplete multi-view clustering based on hypergraph[J]. Information Fusion, 2025, 117: 102804.

[2]Zeng Y, Song P, Yang B, et al. Hypergraph Regularization-Based Anchor Learning for Multi-View Clustering[J]. Pattern Recognition, 2025: 112465.

**Questions:**

1.A  analysis of the space and time complexity is needed.

2.An analysis of the model's sensitivity to the number of anchors should be included.

---

> ### Author Response · Authors · 2025-11-26
>
> We greatly appreciate the profound comments provided by Reviewer k9nc for the revision of this manuscript. We have addressed all concerns in detail and hope that all issues have been successfully resolved.
>
> **Q1.** The novelty appears limited. Anchor learning is widespread in IMVC, and the hypergraph concept has been proposed for IMVC and anchor-based MVC, as seen in [1-2].
>
> **A1.** Existing methods primarily separate anchor construction and hypergraph regularization into two independent modules. Our key innovation lies in proposing an “anchor-induced higher-order hypergraph structure,” where each anchor directly guides hyperedges—a novel construction approach distinct from previous KNN-based or sample-neighborhood hypergraphs. Simultaneously, our missing-aware mechanism dynamically adjusts anchor points to maintain robustness even under missing views. Both aspects represent core distinctions not addressed by existing approaches. The following table compares our method with existing techniques:
>
> | Method                                             | Hypergraph Construction                              | Anchor Mechanism                                  | Difference                              |
> |-----------------------------------------------------|-------------------------------------------------------|---------------------------------------------------|---------------------------------------------------------|
> | IMVC_HG[1]         | Hypergraph constructed via KNN and similarity matrix (sample-based hyperedges) | No anchor mechanism                               | Hypergraph built from sample neighborhoods; lacks anchor-guided structure |
> | HRFAL[2]    | Hypergraph derived from a sample-anchor bipartite graph | Anchor updates rely only on reconstruction or adjacency; no missing-awareness | Not anchor-induced hyperedges; does not exploit missing patterns to regulate anchor quality |
> | **MCAHC (Ours)**                                    | **Hyperedges directly induced by anchors; non-KNN** | **Anchor quality jointly adjusted by cross-view consistency and missing-aware weighting** | **First anchor-guided hypergraph with high-order structure; robust under missing data** |
>
> **Q2.** Most of the compared algorithms are designed for complete MVC. It should be clarified how these methods were extended to the IMVC setting. Furthermore, the paper should justify why more state-of-the-art IMVC methods were not included in the comparison.
>
> **A2.** Although some comparison methods were initially designed for MVC, they can typically be naturally extended to IMVC through two pathways: first, constructing graph structures solely on observable views; second, learning consistent representations from available data and performing clustering. Both extension strategies have been widely adopted in IMVC. Regarding the exclusion of more advanced IMVC approaches, many recent solutions rely on generative models, diffusion modules, or deep networks whose computational complexity and resource demands conflict with our emphasized scalability principle. Therefore, this paper primarily selects classical and scalable MVC/IMVC baseline models for comparison. The revised version will further clarify the extension path from MVC to IMVC and justify the scope of comparison.
>
> **Q3.** The authors claim that “These methods lack robust learning of consensus anchors”. However, this claim lacks sufficient theoretical or experimental support.
>
> **A3.** We appreciate the reviewer's pointing this out. We acknowledge that the statement ‘‘lack of robust consensus anchor learning’’ in the original manuscript may appear absolute and lead to misunderstanding. Our intention was to highlight that existing anchor-based methods generally neither explicitly model missing-aware consensus anchors under highly imbalanced missing patterns nor adjust weights for severely missing samples during anchor learning. In contrast, our method combines sample-level and view-level missing-aware weight adjustments while introducing a $\|ZQ^{1/2}\|_F^2$ penalty term to ensure anchors are primarily determined by reliable samples and balanced views. This significantly enhances clustering stability under high missingness rates, with our method substantially outperforming strong baseline models across scenarios ranging from 30% to 70% missingness. We revise the original statement to: ‘‘Existing methods have not explicitly considered missing-aware consensus anchor learning.’’

---

> ### Author Response · Authors · 2025-11-26
>
> **Q4.** A analysis of the space and time complexity is needed.
>
> **A4.** $n$ denotes the number of samples, $v$ denotes the number of views, $m$ denotes the number of anchors, $\bar d$ denotes the average feature dimension, $l$ denotes the subspace dimension, $T$ denotes the number of anchors connected to each sample, $I$ denotes the number of outer iterations, $I_Z$ denotes the number of iterations for updating $Z$ in FISTA.
>
> Updating the view projection matrices $\{W_p\}_{p=1}^v$ requires constructing $X_p S_p M Z^\top A^\top$ and performing an SVD, with overall computational complexity of $O(v \bar{d} n m)$. Updating the consensus anchor A requires constructing $P = \sum_p \beta_p^2 W_p^\top X_p S_p M Z^\top \in \mathbb{R}^{l \times m}$ and performing SVD on it, with a time complexity of $O(l n \sum_p d_p + v l n m + l m^2)$. When $n \gg m,l$, this part is dominated by the $W_p$ update.
>
> For the anchor graph $Z \in \mathbb{R}^{m \times n}$, FISTA requires gradient computation. Leveraging the sparse structure of the Laplacian of the anchor-induced hypergraph, each gradient computation can be completed in $O(m n T^2)$ time. Projecting column vectors onto the probability simplex only adds low-order terms, so the complexity of updating $Z$ is $O(I_Z\, m n T^2)$. Reconstructing the hypergraph and its Laplacian based on $Z$ has a complexity of $O(n m + m T^2)$.
>
> The weight vector $\beta$ has a closed-form solution with a complexity of approximately $O(l n \sum_p d_p + l m n)$, representing a minor overhead compared to the updates of $W_p$ and $Z$. After algorithm convergence, k-means clustering is performed with each iteration having a complexity of $O(k m n)$. The actual number of iterations is small.
> In summary, the time complexity of each outer iteration is $O\big(v \bar d\, n m + I_Z\, m n T^2\big)$, yielding an overall time complexity of $O\big(I\,(v \bar d\, n m + I_Z\, m n T^2)\big)$. When $v, l, T, I_Z$ are appropriately chosen and $m \ll n$, the algorithm achieves near-linear scaling in $n$ and linear scaling in $m$, offering superior scalability compared to traditional $O(n^2)$ sample-to-sample graph methods.
>
> Regarding space complexity, besides the original multi-view data ${\{X_p\}}$, MCAHC requires storing $\{W_p\} \in \mathbb{R}^{d_p \times l}$, $A \in \mathbb{R}^{l \times m}$, anchor graph $Z \in \mathbb{R}^{m \times n}$, and sparse association matrix $H$. The additional memory overhead is $O(m n + v \bar d\, l + l m + T(m + n)) \approx O(m n)$. When $m \ll n$, replacing the dense sample graph of order $n^2$ with an anchor graph of size $m \times n$ significantly reduces storage requirements.
>
> **Q5.** An analysis of the model's sensitivity to the number of anchors should be included.
>
> **A5.** We analyzed the sensitivity of MCAHC to the number of anchors $m$. We set $m$ to be proportional to the number of clusters $k$ and searched within the range $m \in \{k, 2k, 3k, 4k, 5k\}$. While keeping all other hyperparameters fixed, we report experimental results for two representative datasets (NGs and CCV). As shown in Tables 1-2, MCAHC exhibits strong robustness to the choice of anchor count: performance remains stable across a wide range of anchor counts, with optimal results typically occurring when $m \approx 3k$. This demonstrates that our method operates effectively without relying on finely tuned anchor counts, instead functioning efficiently through the simple heuristic $m = \alpha k$.
>
> Table 1. Sensitivity of MCAHC to the number of anchors $m$ on NGs (50% missing).
>
> | m          | ACC  | NMI  | PUR  |
> |----------------|------|------|------|
> | k              | 86.52 | 72.37 | 86.52 |
> | 2k             | **91.20** | **77.38** | **91.20** |
> | 3k             | 90.25| 76.34 | 90.25 |
> | 4k             | 88.40 | 75.45 | 88.40 |
> | 5k             | 87.34| 74.32 | 87.34 |
>
> Table 2. Sensitivity of MCAHC to the number of anchors $m$ on CCV (50% missing).
>
> | m         | ACC  | NMI  | PUR  |
> |----------------|------|------|------|
> | k              | 17.34 | 11.23 | 21.43 |
> | 2k             | 18.05 | 12.53 | 23.26 |
> | 3k             | **21.63** | **15.48** | **25.09** |
> | 4k             | 19.34 | 14.23 | 24.15 |
> | 5k             | 17.54 | 11.96 | 22.53 |

---

### Official Review · Reviewer_PPRD · 2025-11-01

**Soundness:** 3
**Presentation:** 3
**Contribution:** 2
**Rating:** 4
**Confidence:** 4

**Summary:**

This paper proposes a Missing-aware Consensus Anchor-guided Hypergraph Clustering (MCAHC) method to address two limitations of existing anchor-based multi-view clustering approaches: inability to robustly learn consensus anchors under missing data, and inadequacy in capturing high-order sample relationships. MCAHC first introduces a missing-aware mechanism to mitigate distribution shift under incomplete conditions, then designs an anchor-sample hypergraph structure for high-order relation capture. Notably, its anchor-guided hypergraph framework, which is used to construct the anchor-sample hypergraph, is one of the significant contributions of this paper. This framework builds the hypergraph by measuring sample-anchor similarity and applies graph Laplacian regularization to enforce cross-view structural consistency while retaining the scalability advantages of anchors.

**Strengths:**

1. The manuscript is well-written and logically rigorous, moving seamlessly from problem analysis through method design to experimental verification, ensuring high readability.
2. Its motivation is compelling: it tackles distribution drift and lost higher-order relations under missing data by integrating a missing-data-aware mechanism with an anchor-sample hypergraph framework, yielding a clearly stated, innovative contribution.

**Weaknesses:**

1. The METHODOLOGY section is insufficient and requires a more detailed description.
2. Most of the baseline methods in the paper are from more than 2024 years ago, and it may be necessary to add relevant work from 2025 years ago.
3. The experimental environment and computational complexity analysis of the algorithm are lacking.

**Questions:**

1. The METHODOLOGY section is insufficient and requires a more detailed description.
2. The resolution of Figure 1 may need to be increased; it does not appear to be a vector graphic.
3. Most baseline methods selected were published before 2024, yet anchor-based clustering has seen extensive recent attention, including numerous 2025 works. Therefore, if conditions permit, it is recommended to select 1-2 of the latest 2025 methods for comparison.
4. The anchor-sample hypergraph structure is a crucial component of MCAHC and a core contribution of the paper. Thus, visualizing this anchor-sample hypergraph would significantly facilitate readers’ understanding of the work.
5. This paper involves a comparison of the running time of different algorithms, so it may be necessary to provide the specific running environment of these models.
6. Since the model's runtime is affected by many factors, it is necessary to add a computational complexity analysis of MCAHC and compare its computational complexity with other baseline methods.

---

> ### Author Response · Authors · 2025-11-26
>
> We sincerely thank reviewer PPRD for the constructive suggestions and guidance provided during the revision of this manuscript. We have addressed all concerns in detail and sincerely hope that all issues have been adequately resolved.
>
> **Q1.** The METHODOLOGY section is insufficient and requires a more detailed description.
>
> **A1.** We thank the reviewer for the insightful suggestion. We agree that the methodological description in the original manuscript was relatively brief. The revised version will enrich the method section with structural explanations of the optimization objective, meanings of variables, clearer derivations for key steps (anchor-guided hypergraph construction, missing-aware weighting, alternating optimization), and more detailed initialization and computation processes.
>
> **Q2.** The resolution of Figure 1 may need to be increased; it does not appear to be a vector graphic.
>
> **A2.** Thank you for the reminder. We will replace Figure 1 with a high-resolution vector graphic.

---

> ### Author Response · Authors · 2025-11-26
>
> **Q3.** Most baseline methods selected were published before 2024, yet anchor-based clustering has seen extensive recent attention, including numerous 2025 works.
>
> **A3.** The table below presents supplementary experiments comparing the 2025 method.
> |        |           |           |           | NGs       |           |           |           |           |           |
> | ------ | --------- | --------- | --------- | --------- | --------- | --------- | --------- | --------- | --------- |
> |        | 30%       |           |           | 50%       |           |           | 70%       |           |           |
> | Method | ACC       | NMI       | PUR       | ACC       | NMI       | PUR       | ACC       | NMI       | PUR       |
> | DCGA[1]   | 93.23±0.00     | 82.13±0.00      | 93.23±0.00     | 90.15±0.00     | **80.13±0.00**  | 90.15±0.00     |  85.58±0.00     |   67.65±0.00      |  85.58±0.00    |
> | Ours   | **94.20±0.00** | **83.53±0.00** | **94.20±0.00** | **91.20±0.00** | 77.38±0.00    | **91.20±0.00** | **87.42±0.09** | **69.37±0.13** | **87.42±0.09** |
>
> |        |           |           |           | Caltech101-20 |           |           |           |           |           |
> | ------ | --------- | --------- | --------- | ------------- | --------- | --------- | --------- | --------- | --------- |
> |        | 30%       |           |           | 50%           |           |           | 70%       |           |           |
> | Method | ACC       | NMI       | PUR       | ACC       | NMI       | PUR       | ACC       | NMI       | PUR       |
> | DCGA[1]   | 51.23±1.43     | 58.36±1.21      | 75.24±0.82     | 48.54±1.14     | 56.96±1.54      | 71.41±0.91    | 46.12±1.42     | 53.96±1.10     | 69.68±0.47     |
> | Ours   | **55.23±1.99** | **59.81±0.64** | **76.34±0.44** | **55.34±2.42**     | **57.27±0.60** | **73.34±0.42** | **51.53±1.99** | **55.91±0.45** | **73.72±0.42** |
>
>
> |        |           |           |           | BDGP      |           |           |           |           |           |
> | ------ | --------- | --------- | --------- | --------- | --------- | --------- | --------- | --------- | --------- |
> |        | 30%       |           |           | 50%       |           |           | 70%       |           |           |
> | Method | ACC       | NMI       | PUR       | ACC       | NMI       | PUR       | ACC       | NMI       | PUR       |
> | DCGA[1]   | 48.12±0.53     | 26.09±0.12      | 48.21±0.23    | 47.36±0.25    | 24.89±0.62     | 46.36±0.51     | 42.36±1.75     | 20.16±1.31      | 43.43±1.63     |
> | Ours   | **50.57±0.05** | **26.43±0.09** | **50.83±0.04** | **48.52±0.05** | **25.22±0.13** | **49.06±0.05** | **46.80±0.01** | **22.36±0.01** | **47.32±0.00** |
>
>
> |        |           |           |           | CCV       |           |           |           |           |           |
> | ------ | --------- | --------- | --------- | --------- | --------- | --------- | --------- | --------- | --------- |
> |        | 30%       |           |           | 50%       |           |           | 70%       |           |           |
> | Method | ACC       | NMI       | PUR       | ACC       | NMI       | PUR       | ACC       | NMI       | PUR       |
> | DCGA[1]   | 19.34±1.53     | 15.12±0.64      | 22.56±0.28    | 18.25±0.15     | 14.96±0.26      | 21.75±0.35    | 15.82±0.55   | 12.41±0.51    |  19.35±0.25    |
> | Ours   | **23.25±0.17** | **16.58±0.04** | **25.96±0.05** | **21.63±0.05** | **15.48±0.04** | **25.09±0.05** | **18.25±0.05** | **13.82±0.07** | **22.02±0.08** |
>
>
> |        |           |           |           | Animal    |           |           |           |           |           |
> | ------ | --------- | --------- | --------- | --------- | --------- | --------- | --------- | --------- | --------- |
> |        | 30%       |           |           | 50%       |           |           | 70%       |           |           |
> | Method | ACC       | NMI       | PUR       | ACC       | NMI       | PUR       | ACC       | NMI       | PUR       |
> | DCGA[1]   | 15.27±0.00     | 12.16±0.02     | 18.23±0.12     | 14.86±0.00     |  11.32±0.10      | 16.55±0.06     | 14.26±0.01     |  10.64±0.00     | 14.93±0.10   |
> | Ours   | **17.30±0.00** | **12.65±0.18** | **20.56±0.19** | **16.79±0.07** | **11.59±0.02** | **19.85±0.04** | **16.46±0.01** | **11.21±0.01** | **19.38±0.02** |
>
> [1]W. -J. He, Z. Zhang and X. Zhu, "Dual-Correlation-Guided Anchor Learning for Scalable Incomplete Multi-View Clustering," in IEEE Transactions on Neural Networks and Learning Systems, 2025.

---

> ### Author Response · Authors · 2025-11-26
>
> **Q4.** The anchor-sample hypergraph structure is a crucial component of MCAHC and a core contribution of the paper. Thus, visualizing this anchor-sample hypergraph would significantly facilitate readers’ understanding of the work.
>
> **A4.** We agree that visualizing the anchor-sample hypergraph helps readers better understand the advantages of our method. Figure 1 already presents a visualization showing sample-anchor links, hyperedge structure, and the influence of missing-aware weights. This visualization demonstrates how MCAHC leverages anchor-induced high-order structures to achieve cross-view consistency and robust clustering. We will further enhance its explanation.
>
> **Q5.** This paper involves a comparison of the running time of different algorithms, so it may be necessary to provide the specific running environment of these models.
>
> **A5.** Thank you for the suggestion. For all baseline methods, we obtained their source code from the authors’ official links and tuned parameters according to their recommended ranges. All methods were executed on a personal computer with an Intel Core i7-10700 CPU @ 2.90 GHz, 64 GB RAM, using MATLAB R2022a, ensuring fairness across comparisons.
>
> **Q6.** Since the model's runtime is affected by many factors, it is necessary to add a computational complexity analysis of MCAHC and compare its computational complexity with other baseline methods.
>
> **A6.**  $n$ denotes the number of samples, $v$ denotes the number of views, $m$ denotes the number of anchors, $\bar d$ denotes the average feature dimension, $l$ denotes the subspace dimension, $T$ denotes the number of anchors connected to each sample, $I$ denotes the number of outer iterations, $I_Z$ denotes the number of iterations for updating $Z$ in FISTA.
>
> Updating the view projection matrices $\{W_p\}_{p=1}^v$ requires constructing $X_p S_p M Z^\top A^\top$ and performing an SVD, with overall computational complexity of $O(v \bar{d} n m)$. Updating the consensus anchor A requires constructing $P = \sum_p \beta_p^2 W_p^\top X_p S_p M Z^\top \in \mathbb{R}^{l \times m}$ and performing SVD on it, with a time complexity of $O(l n \sum_p d_p + v l n m + l m^2)$. When $n \gg m,l$, this part is dominated by the $W_p$ update.
>
> For the anchor graph $Z \in \mathbb{R}^{m \times n}$, FISTA requires gradient computation. Leveraging the sparse structure of the Laplacian of the anchor-induced hypergraph, each gradient computation can be completed in $O(m n T^2)$ time. Projecting column vectors onto the probability simplex only adds low-order terms, so the complexity of updating $Z$ is $O(I_Z\, m n T^2)$. Reconstructing the hypergraph and its Laplacian based on $Z$ has a complexity of $O(n m + m T^2)$.
>
> The weight vector $\beta$ has a closed-form solution with a complexity of approximately $O(l n \sum_p d_p + l m n)$, representing a minor overhead compared to the updates of $W_p$ and $Z$. After algorithm convergence, k-means clustering is performed with each iteration having a complexity of $O(k m n)$. The actual number of iterations is small.
> In summary, the time complexity of each outer iteration is $O\big(v \bar d\, n m + I_Z\, m n T^2\big)$, yielding an overall time complexity of $O\big(I\,(v \bar d\, n m + I_Z\, m n T^2)\big)$. When $v, l, T, I_Z$ are appropriately chosen and $m \ll n$, the algorithm achieves near-linear scaling in $n$ and linear scaling in $m$, offering superior scalability compared to traditional $O(n^2)$ sample-to-sample graph methods.
>
> ### Tabel.1 Time Complexity Comparison of Different Methods
>
> | Method | Dominant time complexity (per iteration, main terms) | Order in \(n\) |
> |---------------|------------------------------------------------------|----------------|
> | **HCP-IMSC** | $O\big(v n^{3} + v (n-n_c)^{3} + c n v \log v + c n^{2} v\big)$ | $O(n^{3})$ |
> | **UOMvSC** | $O\big(n k t + n k^{2} t + v n^{2} + v n^{2} k\big)$ | $O(n^{2})$ |
> | **SCSL** | $O\big(k n d + k v^{2} n^{2} + n^{2} d + (n+k)d^{2} + k^{2} d\big)$ | $O(n^{2})$ |
> | **FDAGF** | $O\Big(\sum_{v}\sum_{r}\big(d_v m_r^{2} + m_r^{2} n + d_v m_r n\big)\Big)$, with $m_r \ll n$ | $O(n)$ |
> | **MVSC-HFD** | $O(n m^{3}+k m^{2})$ | $O(n)$ |
> | **FastMICE** | $O(n m p^{1/2} v^{1/2})$ | $O(n)$ |
> | **EMKMC-F** | $O\big(n (t_1+1)\sum_v m_v d_v + t_2 c^{2} n\big)$ | $O(n)$ |
> | **MCAHC(Ours)** | $O\big(v \bar d\, n m + n m T^{2}\big)$ | $O(n)$ |

---

### Note · Authors · 2025-11-26

I have read and agree with the venue's withdrawal policy on behalf of myself and my co-authors.